# Beyond Autoregression: Discrete Diffusion for Complex Reasoning and Planning

**Jiacheng Ye**[1]**, Jiahui Gao**[2] **Shansan Gong**[1]**, Lin Zheng**[1]
**Xin Jiang**[2]**, Zhenguo Li**[2]**, Lingpeng Kong**[1]
[1] The University of Hong Kong     [2] Huawei Noah's Ark Lab
carsonye@connect.hku.hk

## Abstract

Autoregressive language models, despite their impressive capabilities, struggle with complex reasoning and long-term planning tasks. We introduce discrete diffusion models as a novel solution to these challenges. Through the lens of subgoal imbalance, we demonstrate how diffusion models effectively learn difficult subgoals that elude autoregressive approaches. We propose Multi-Granularity Diffusion Modeling (MGDM), which prioritizes subgoals based on difficulty during learning. On complex tasks like Countdown, Sudoku, and Boolean Satisfiability Problems, MGDM significantly outperforms autoregressive models without using search techniques. For instance, MGDM achieves 91.5% and 100% accuracy on Countdown and Sudoku, respectively, compared to 45.8% and 20.7% for autoregressive models. Our work highlights the potential of diffusion-based approaches in advancing AI capabilities for sophisticated language understanding and problem-solving tasks. All associated codes are available at https://github.com/HKUNLP/diffusion-vs-ar.

## 1 Introduction

In recent years, autoregressive language models (LMs; Bengio et al. 2000) have dominated the landscape of natural language processing and artificial intelligence. Empowered by scaling laws (Kaplan et al., 2020), these models have demonstrated impressive performance across various applications (OpenAI, 2022; Achiam et al., 2023; Anthropic, 2023; Team et al., 2023, *inter alia*). However, this apparent success masks significant limitations that are becoming increasingly evident. Autoregressive models inherently struggle with tasks requiring complex reasoning, long-term planning, and maintaining global coherence (Bubeck et al., 2023; Valmeekam et al., 2023; 2024; Dziri et al., 2024; Kambhampati et al., 2024). These shortcomings represent substantial challenges in developing AI systems capable of robust problem-solving and adaptable cognition (Wu et al., 2022; Zhao et al., 2023; Trinh et al., 2024; Yao et al., 2023; Shinn et al., 2024, *inter alia*). While autoregressive approaches have driven considerable progress, their limitations suggest that they may not be the optimal solution for all aspects of machine intelligence. As the field evolves, it becomes increasingly important to explore alternative paradigms that can address these inherent drawbacks and potentially offer new avenues for advancement in AI capabilities.

In response to these limitations, recent research has focused on addressing the inherent constraints of autoregressive models. Various strategies have been explored, including the integration of search algorithms at inference (Yao et al., 2024; Besta et al., 2024) and the incorporation of backtracking supervision during training (Lehnert et al., 2024; Gandhi et al., 2024). However, these approaches are not without their own drawbacks: the former often incurs significant computational costs, while the latter frequently results in verbose inputs and suboptimal performance.

To address this challenge, we argue for a fundamentally different modeling approach: discrete diffusion models. While most contemporary language models are autoregressive, diffusion-based models have become predominant in image (Dhariwal & Nichol, 2021; Rombach et al., 2022; Peebles & Xie, 2023) and video domains (Ho et al., 2022; Wu et al., 2023a; Brooks et al., 2024). Diffusion models are also gaining traction in various other applications, such as protein desiging (Xu et al., 2022; Hoogeboom et al., 2022b; Corso et al., 2023) and planning in reinforcement learning (Jan-

ner et al., 2022; Ajay et al., 2022; Chi et al., 2023). In this work, we reveal that discrete diffusion models demonstrate significantly superior performance compared to the autoregressive counterparts, particularly in tasks requiring complex planning and reasoning.

To substantiate this argument, we first examine the problem through the lens of *subgoal imbalance* (§3.1). We present both theoretical and empirical evidence via a synthetic planning task (Figure 1) to illustrate why autoregressive models struggle with these types of problems, often achieving near-random performance. In contrast, we demonstrate how diffusion models effectively learn the subgoals that challenge autoregressive models (§3.2). The key insight lies in the training objective of diffusion models, where difficult subgoals are decomposed into a diverse range of interrelated views within a multi-view learning framework (Xu et al., 2013). Each of these views is more manageable, resulting in an overall easier and more effective learning process.

Building upon these insights, we propose a natural extension to current discrete diffusion models, which we term multi-granularity diffusion modeling (MGDM; §3.3). This approach prioritizes different subgoals based on their difficulty during the learning process, leading to more effective learning outcomes and faster convergence.

In our experimental evaluation (§4), we focus on substantially more complex problem-solving tasks, such as Countdown (Gandhi et al., 2024) and Sudoku (Garns, 1979). These tasks demand extensive planning over a large number of combinations and pose challenges even for commercial Large Language Models (e.g., GPT-4 Achiam et al. 2023). Notably, without employing any search techniques, MGDM achieves 91.5% and 100% accuracy on Countdown and Sudoku respectively, while its autoregressive counterpart only solves 45.8% and 20.7% of the problems. Additionally, we conduct experiments on the Boolean Satisfiability Problem (SAT), an NP-complete problem (Cook, 1971) that represents a wide range of constraint satisfaction problems. Our model exhibits superior performance in solving SAT problems with higher accuracy compared to the autoregressive alternative, particularly when dealing with an increased number of variables and constraints. Through this systematic exploration, we aim to demonstrate the potential advantages of diffusion-based approaches in addressing sophisticated language understanding and generation challenges.

## 2 BACKGROUND

### 2.1 AUTO-REGRESSIVE MODELING

Let $\mathbf{x} := (\boldsymbol{x}_1, \ldots, \boldsymbol{x}_N)$ denote a sequence drawn from a data distribution $q(\mathbf{x})$. For decades, it has been common to factorize the joint probabilities of a sequence of tokens as the product of conditional probabilities (Jelinek, 1980; Bengio et al., 2000):

$$p_{\boldsymbol{\theta}}(\mathbf{x}) = p_{\boldsymbol{\theta}}(\boldsymbol{x}_1) \prod_{n=2}^{N} p_{\boldsymbol{\theta}}(\boldsymbol{x}_n \mid \boldsymbol{x}_{1:n-1}), \tag{1}$$

where $\boldsymbol{\theta}$ parameterizes the model distribution and $\boldsymbol{x}_{1:n-1} := \boldsymbol{x}_1, \ldots, \boldsymbol{x}_{n-1}$. In order to optimize the generative model $p_{\boldsymbol{\theta}}(\mathbf{x})$ to fit the data distribution $q(\mathbf{x})$, we optimize the negative log-likelihood:

$$L_{\text{AR}} = -\mathbb{E}_{q(\mathbf{x})} \log p_{\boldsymbol{\theta}}(\mathbf{x}) = -\mathbb{E}_{q(\mathbf{x})} \sum_{n=1}^{N} \log p_{\boldsymbol{\theta}}(\boldsymbol{x}_n \mid \boldsymbol{x}_{1:n-1}). \tag{2}$$

### 2.2 DISCRETE DIFFUSION MODELING

Discrete diffusion models (Sohl-Dickstein et al., 2015; Hoogeboom et al., 2021; Austin et al., 2021) are a class of latent variable models characterized by a forward noising process and a learned reverse denoising process. The forward process $q(\mathbf{x}_{1:T}|\mathbf{x}_0) = \prod_{t=1}^{T} q(\mathbf{x}_t|\mathbf{x}_{t-1})$ corrupts the original data $\mathbf{x}_0 := \mathbf{x}$ into a sequence of increasingly noisy latent variables $\mathbf{x}_{1:T} := \mathbf{x}_1, \ldots, \mathbf{x}_T$. The backward process learns to gradually denoise the latent variables to the data distribution given by:

$$p_{\boldsymbol{\theta}}(\mathbf{x}) = \sum_{\mathbf{x}_{1:T} \sim q} p(\mathbf{x}_T) \prod_{t=1}^{T} p_{\boldsymbol{\theta}}(\mathbf{x}_{t-1}|\mathbf{x}_t). \tag{3}$$

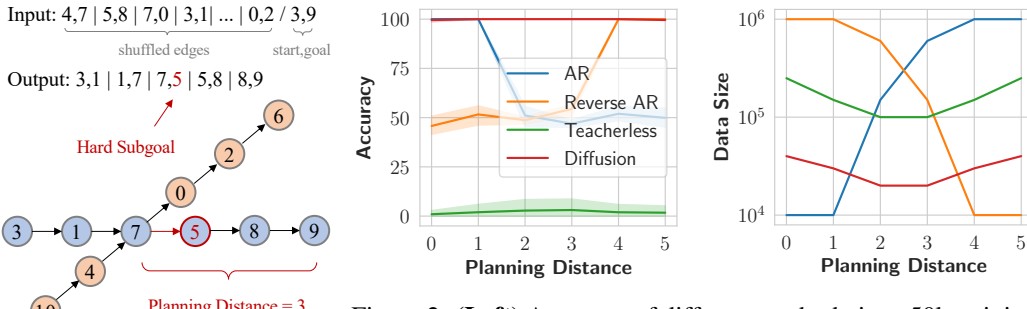

Figure 1: The planning task.

Figure 2: **(Left)** Accuracy of different method given 50k training data. **(Right)** Minimum data size required to solve (i.e., accuracy above 90%) subgoal at each planning distance.

Due to the intractable marginalization, we typically optimize a variational upper bound on the negative log-likelihood:

$$L_{\mathrm{DM}} = \mathbb{E}_{q(\mathbf{x}_0)}\Bigg[ \underbrace{D_{\mathrm{KL}}[q(\mathbf{x}_T|\mathbf{x}_0)||p(\mathbf{x}_T)]}_{L_T} + \sum_{t=2}^{T} \underbrace{\mathbb{E}_{q(\mathbf{x}_t|\mathbf{x}_0)}\big[D_{\mathrm{KL}}[q(\mathbf{x}_{t-1}|\mathbf{x}_t,\mathbf{x}_0)||p_{\boldsymbol{\theta}}(\mathbf{x}_{t-1}|\mathbf{x}_t)]\big]}_{L_{t-1}}$$
$$\underbrace{-\mathbb{E}_{q(\mathbf{x}_1|\mathbf{x}_0)}[\log p_{\boldsymbol{\theta}}(\mathbf{x}_0|\mathbf{x}_1)]}_{L_0}\Bigg], \tag{4}$$

where $L_T$ is a constant when one uses a fixed prior $p(\mathbf{x}_T)$. By defining both the forward and backward distribution as categorical distribution, e.g., $q(\mathbf{x}_t|\mathbf{x}_{t-1}) = \mathrm{Cat}(\mathbf{x}_t; \boldsymbol{p} = \boldsymbol{Q}_t^\top \mathbf{x}_{t-1})$ where $\boldsymbol{Q}_t$ is a pre-defined $K \times K$ transition matrix and $K$ is the size of categories, and $p_{\boldsymbol{\theta}}(\mathbf{x}_{t-1}|\mathbf{x}_t) = q(\mathbf{x}_{t-1}|\mathbf{x}_t, f(\mathbf{x}_t; \boldsymbol{\theta}))$, the forward process posterior $q(\mathbf{x}_{t-1}|\mathbf{x}_t, \mathbf{x}_0)$ and each KL term can be calculated analytically (Hoogeboom et al., 2021; Austin et al., 2021).

## 3  SUBGOAL IMBALANCE AND MULTI-GRANULARITY DIFFUSION MODELS

In this section, we employ a motivation example (§3.1) to elucidate the challenges faced by autoregressive models in specific scenarios. Through this analysis, we introduce the concept of *subgoal imbalance*—wherein some subgoals are inherently more difficult than others—which offers insights into these difficulties. We then extend our discussion in §3.2 to examine how diffusion models can more effectively address and learn these *hard subgoals*, effectively overcoming the limitations observed in autoregressive approaches. We finally propose Multi-Granularity Diffusion Modeling (MGDM; §3.3) as a natural extension of discrete diffusion models to better address these challenges and improve performance on complex tasks requiring planning and reasoning.

### 3.1  SUBGOAL IMBALANCE IN AUTOREGRESSIVE AND DIFFUSION MODELING

We designed a simple planning task to serve as our running example. Consider the example in Figure 1, where the input for the task consists of a set of shuffled edges from the graph shown below. At the end of the input sequence, the start and goal nodes are specified to indicate the path the model needs to find. The objective of this task is to identify the correct path in the graph and output its constituent edges. The complexity of this problem arises from distracting factors (highlighted in orange) that potentially mislead the path selection. For instance, at node 7, with the goal being node 9, the model must plan over a distance of 3 nodes to determine that the correct next choice should be node 5 rather than 0. We define this span as the Planning Distance (PD), a parameter adjustable in our synthetic task data. Intuitively, as the PD increases, the model faces greater difficulty in learning to determine the correct subsequent node. We formalize this intuition as *subgoal imbalance*.

**Proposition 1** (*Subgoal imbalance due to the unknown data distribution* $q(\mathbf{x})$) *Given the data* $\mathbf{x}$ *sampled from an unknown data distribution* $q(\mathbf{x})$, *the difficulty of learning each subgoal* $\boldsymbol{x}_n$ *can differ significantly based on how we parametrize the model distribution, and some subgoals may require substantially more data to learn or may even be infeasible to learn.*

**Subgoal imbalance in autoregressive modeling.** Given the data $\mathbf{x}$ sampled from an unknown data distribution $q(\mathbf{x})$, the naive autoregressive modeling parametrizes the model distribution $p_{\boldsymbol{\theta}}(\mathbf{x})$ as $p_{\boldsymbol{\theta}}(\boldsymbol{x}_1) \prod_{n=2}^{N} p_{\boldsymbol{\theta}}(\boldsymbol{x}_n \mid \boldsymbol{x}_{1:n-1})$, the difficulty of learning each subgoal $\boldsymbol{x}_n$ can differ significantly as given only the left context, and some subgoals may require substantially more data to learn or may even be infeasible to learn.

**Setup.** We synthesize the data with only one distracting path. We randomize node numbers in $[0, 10]$ and the intersection positions in $[0, 5]$. We further designed this task to be symmetric, ensuring that simply training with reversed output, as suggested by Bachmann & Nagarajan (2024), cannot solve subgoals with all PDs. For comparison, we include Auto-regressive (AR), reverse AR (Bachmann & Nagarajan, 2024), and teacherless training (Monea et al., 2023; Bachmann & Nagarajan, 2024), which can be seen as a lookahead method that produce all target tokens from the source input, and our proposed diffusion model (detailed in §3.2). For all the models, we keep the model architecture fixed as the same 3-layer Transformer with approximately 6M parameters. More details can be found in Appendix §C.

**Discussion.** We examine the performance of all the models in two scenarios. In the first scenario, we generate a fixed number of 50k instances with mixed planning distance. We plot the accuracy on the held-out evaluation set for each model in the left figure of Figure 2. Our findings indicate that autoregressive models (AR and Reverse AR) are only effective in solving cases where the PD equals 0 or 1 (or equivalently, 5 and 4 in the reverse setting). Due to the aforementioned subgoal imbalance phenomenon, when PD is less than 2, the task barely involves any planning, allowing models to simply copy from the input with ease. However, for larger PDs, AR models barely outperform random guessing. Teacherless training fails to adequately fit the training data, resulting in the production of illegal paths. In contrast, our diffusion model achieves perfect accuracy across all PD values.

In the second scenario, we investigate whether the challenging subgoals can be naturally resolved through data or model scaling, akin to the success observed in large language models (Kaplan et al., 2020; Wei et al., 2022a). To investigate this question, we gradually increase the size of the dataset for each model with different PDs and plot the minimum data size required to solve the subgoal in the right figure of Figure 2. We find that the autoregressive models (AR and Reverse AR) can learn the easy cases of PD equal to 0 and 1 (or equivalently, 5 and 4 in the reverse setting) with only 10k data points. However, exponentially larger amounts of data are required to address increasingly challenging subgoals. Both teacherless training and diffusion models exhibit a similar U-shaped curve in their performance. This similarity can be attributed to the fact that teacherless training can be conceptualized as a special case of diffusion without an iterative noising and denoising process. In these models, solving edge PDs necessitates slightly more data. We hypothesize that this phenomenon occurs because the distance to other positions is shorter from the middle position (i.e., higher closeness centrality), thus providing the middle position with more nearby tokens to aid in prediction. Overall, autoregressive models require significantly more data to address all PDs compared to diffusion models, highlighting their relative data inefficiency.

In addition to our previous experiments, we conducted a series of tests to examine the effect of increasing the parameter count in autoregressive models while maintaining a fixed dataset size of 50,000 instances. Our findings reveal that scaling the original 6 million parameter model to 85 million, 303 million, and 1.5 billion parameters fails to resolve all PDs. Only upon fine-tuning a substantially larger model, specifically the LLaMA 7B model (Touvron et al., 2023), did we observe successful resolution of all PD subgoals.

## 3.2 EFFECTIVE HARD SUBGOAL LEARNING IN DIFFUSION MODELING

These experiments collectively indicate that diffusion models are significantly more effective in learning challenging subgoals arising from subgoal imbalance. To elucidate why diffusion models exhibit this superior capability, we first establish a connection between autoregressive (AR) models and diffusion models by reformulating Equation (4). Instead of evaluating the KL divergence between two complicated categoricals (Hoogeboom et al., 2021), we consider discrete diffusion with absorbing state and simplify it as the weighted cross-entropy losses (Austin et al., 2021; Zheng et al.,

2023; Shi et al., 2024; Sahoo et al., 2024):

$$D_{\mathrm{KL}}[q(\mathbf{x}_{t-1}|\mathbf{x}_t,\mathbf{x}_0)||p_{\boldsymbol{\theta}}(\mathbf{x}_{t-1}|\mathbf{x}_t) = -w(t)\sum_{n=1}^{N}\mathbf{1}_{\mathbf{x}_{t,n}\neq\mathbf{x}_{0,n}}\mathbf{x}_{0,n}^{\top}\log f(\mathbf{x}_t;\boldsymbol{\theta})_n, \quad (5)$$

where $w(t) = \frac{\alpha_{t-1}-\alpha_t}{1-\alpha_t} \in (0,1]$ is a time-dependent reweighting term which places higher weight when $t$ approaching 0. We then rewrite Equation (4) as:

$$L_{\mathrm{DM}} = \mathbb{E}_{q(\mathbf{x}_0)}\sum_{n=1}^{N}\underbrace{\sum_{t=1}^{T}w(t)\mathbb{E}_{q(\mathbf{x}_t|\mathbf{x}_0)}u(\mathbf{x}_0,\mathbf{x}_t,n;\boldsymbol{\theta})}_{-\log p_{\mathrm{DM}}(\boldsymbol{x}_n|\boldsymbol{x}_{\neq n})}, \quad (6)$$

where $u(\mathbf{x}_0,\mathbf{x}_t,n;\boldsymbol{\theta}) := -\mathbf{1}_{\mathbf{x}_{t,n}\neq\mathbf{x}_{0,n}}\mathbf{x}_{0,n}^{\top}\log f(\mathbf{x}_t;\boldsymbol{\theta})_n$ is the cross entropy loss on token $n$.

We can now systematically compare the losses of autoregressive (AR) and diffusion models (DM), specifically $-\log p_{\mathrm{AR}}(\boldsymbol{x}_n \mid \boldsymbol{x}_{1:n-1})$ and $-\log p_{\mathrm{DM}}(\boldsymbol{x}_n \mid \boldsymbol{x}_{\neq n})$, as expressed in Equations (2) and (6), respectively. In Figure 3, we examine a specific hard subgoal with Planning Distance (PD) equals 3 in both model types. The loss levels of AR and diffusion models are depicted using blue and red lines, respectively. The overall loss $-\log p_{\mathrm{DM}}(\boldsymbol{x}_n \mid \boldsymbol{x}_{\neq n})$ in the diffusion model remains relatively low compared to its autoregressive counterpart $-\log p_{\mathrm{AR}}(\boldsymbol{x}_n \mid \boldsymbol{x}_{1:n-1})$, corroborating the superior performance of the diffusion model on these challenging subgoals in our experiments.

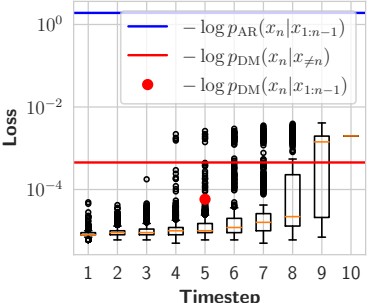

Figure 3: Loss for a specific hard subgoal, i.e., PD=3, in Diffusion and AR modeling. We also show the unweighted loss $u(\mathbf{x}_0,\mathbf{x}_t,n;\boldsymbol{\theta})$ at different timestep $t$ and context $\mathbf{x}_t$ in diffusion modeling.

Further analysis of the unweighted loss $u(\mathbf{x}_0,\mathbf{x}_t,n;\boldsymbol{\theta})$ in the diffusion model, based on 1,000 samples of $\mathbf{x}_t \sim q(\mathbf{x}_t|\mathbf{x}_0)$, reveals a clear trend: as the number of timesteps increases, resulting in more noise in $\mathbf{x}_t$, objectives in smaller timesteps (i.e., recovery from less noisy data) become significantly easier to learn. From a multi-view learning perspective (Xu et al., 2013), each $\mathbf{x}_t$ can be interpreted as a distinct view of $\mathbf{x}_0$, where each view provides different information about $\mathbf{x}_0$. In the diffusion process, by exploring the consistency and complementary properties of different views offered by a diverse range of interrelated objectives $u(\mathbf{x}_0,\mathbf{x}_t,n;\boldsymbol{\theta})$, our findings suggest that objectives challenging to learn in AR models become more effective, promising, and exhibit better generalization in diffusion models.

This phenomenon is particularly evident when examining scenarios where mask noise is applied to positions after the hard token, i.e., $\mathbf{x}_t = \boldsymbol{x}_{1:n-1}$, where the diffusion model learns the hard subgoal similarly to AR models. We plot this loss as $-\log p_{\mathrm{DM}}(\boldsymbol{x}_n \mid \boldsymbol{x}_{1:n-1})$ in the figure. Unlike in the AR model, where this learning is consistently difficult, in diffusion models, this challenging subgoal is addressed at a much more manageable level during the learning process.

## 3.3 MULTI-GRANULARITY DIFFUSION MODELING

These observations provide valuable insights, i.e., diffusion modeling builds on a diverse range of interrelated views from the data $\mathbf{x}_0$ to handle a challenging subgoal. To handle multiple challenging subgoals in real data, we should prioritize different subgoals based on their difficulty during the learning process to achieve more effective learning outcomes and faster convergence, and this naturally translates to prioritizing difficult views as the learning of a subgoal depends on learning interrelated views related to it. Building on this, we propose the multi-granularity diffusion model as a natural extension of the discrete diffusion model.

In practice, to optimize Equation (6), we typically employ Monte Carlo sampling, which results in:

$$L_{\mathrm{DM}} = \sum_{n=1}^{N}\sum_{t=1}^{T}w(t)u(\mathbf{x}_0,\mathbf{x}_t,n;\boldsymbol{\theta}). \quad (7)$$

For a sequence of length $N$, the probability of sampling the same $\mathbf{x}_t$ in AR is 1. However, in diffusion, this probability reduces to $1/C_{N-1}^{t(N-1)/T}$ due to the randomness in sampling $\mathbf{x}_t$, potentially reducing the training efficiency of diffusion models. We note that Equation (7) employs a sequence-level reweighting term $w(t)$ to indicate the importance of $\mathbf{x}_t$. However, individual tokens within the sequence, given their imbalanced difficulties, are not properly reweighted. To address this, we propose multi-granularity diffusion modeling (MGDM), which introduces an additional token-level reweighting mechanism to enhance training efficiency:

$$L_{\text{MGDM}} = \sum_{n=1}^{N} \sum_{t=1}^{T} w(t) v(\mathbf{x}_{t,n}) u(\mathbf{x}_0, \mathbf{x}_t, n; \boldsymbol{\theta}), \tag{8}$$

where $v(\mathbf{x}_{t,n}) = \alpha(1 - \exp(-u(\cdot)))^\beta$ is the adaptive token-level reweighting term. Setting $\beta > 0$ reduces the relative loss for easy tokens while emphasizing harder tokens, and $\alpha$ is used to control the relative reweighting magnitude. For inference, we employ an easy-first TopK decoding strategy, which has demonstrated superior performance compared to the random decoding method used by Austin et al. (2021). This finding aligns with similar observations documented in prior studies (Savinov et al., 2021; Zheng et al., 2023). We provide a detailed derivation and algorithm of the training and inference process in Appendix §A and §B, respectively.

## 4 EXPERIMENTS

In Section §3.1 we show our model works well on a straightforward planning task with only one hard subgoal. However, it is important to note that real-world scenarios often involve instances with multiple challenging subgoals. In this section, we aim to assess the performance of our model in tackling three considerably more complex problem-solving tasks that necessitate deliberate planning. Detailed experimental setup can be found in Appendix §C.

### 4.1 COUNTDOWN

Countdown (Countdown, 2024) is a mathematical reasoning challenge and is a generalized version of the game of 24, which even advanced models such as GPT-4 struggle with (Yao et al., 2024). The goal of Countdown is to use the given numbers and arithmetic operations ($+ - */$) to obtain a target number. For example, given 4 numbers "97,38,3,17" and a target number "14", a step-by-step solution is "97-38=59,59-17=42,42/3=14".

**Setup.** We follow Gandhi et al. (2024) to generate 500k problems with target numbers ranging from 10 to 100 and randomly hold out 10% of the targets for 'out-of-distribution' evaluation. We consider three subtasks with increasing complexity by varying the number of input digits in {3,4,5}. Given that search-augmented prompting approaches (Yao et al., 2024) have recently been employed to address the limitations of AR, we also compare with such approaches by training on Countdown 4 and evaluating on the same game of 24 test set as Yao et al. (2024).

Table 1: Results on the Countdown (CD) task with increasing complexity.

|  | Params | CD 3 | CD 4 | CD 5 |
|---|---|---|---|---|
| *Autoregressive* | | | | |
|  | 6M | 94.1 | 31.9 | 4.3 |
| GPT-2 Scratch | 85M | 95.9 | 45.8 | 5.1 |
|  | 303M | 96.4 | 41.3 | 4.5 |
| Stream-of-Search | 250M | - | 54.2 | - |
| LLaMA | 7B | 95.7 | 41.1 | 6.7 |
|  | 13B | 96.5 | 51.1 | 7.4 |
| *Diffusion* | | | | |
| VDM | 85M | 99.1 | 73.4 | 16.3 |
| D3PM | 85M | 99.4 | 83.1 | 27.6 |
| RDM | 85M | 99.5 | 87.0 | 45.8 |
|  | 6M | 98.1 | 52.0 | 27.0 |
| **MGDM (Ours)** | 85M | 99.5 | **91.5** | **46.6** |
|  | 303M | **99.9** | 88.3 | 39.0 |

**Baselines.** Our primary comparison involves autoregressive models trained from scratch, employing the GPT-2 architecture (Radford et al., 2019) with parameter sizes ranging from 6M, 85M, and 303M (denoted as GPT-2 Scratch). We also include larger pre-trained AR models LLaMA (Touvron et al., 2023) with sizes 7B and 13B. These models are fine-tuned using the same dataset. In addition, we compare with Stream-of-Search (Gandhi et al., 2024), which augments the dataset with search trajectory such that the AR model can be taught to search. Furthermore, we compare with several existing diffusion models,

both continuous models VDM (Kingma et al., 2021) and discrete models D3PM Austin et al. (2021) and RDM (Zheng et al., 2023). By default, we use the absorbing noise for discrete diffusion as it significantly outperforms the multinomial one (Austin et al., 2021; Zheng et al., 2023). Finally, we consider in-context learning (Brown et al., 2020; Wu et al., 2023c; Ye et al., 2023a) based on GPT-4, including vanilla input-output (IO), chain-of-thought (CoT; Wei et al. 2022b), CoT with Self-consistency (CoT-SC; Wang et al. 2023) and Tree-of-thought (ToT; Yao et al. 2024). We use 5 in-context examples following Yao et al. 2024.

**Results on Countdown.** As shown in Table 1, diffusion-based approaches demonstrate superior performance across all three Countdown tasks compared to autoregressive models, especially as the complexity of the tasks increases. We have several key findings based on the result. Firstly, the 6M diffusion model outperforms both the 303M GPT-2 model trained from scratch and the pretrained 13B LLaMA model, indicating that the modeling approach sometimes outweighs the sheer number of parameters. Secondly, while training with search trajectory supervision (Stream-of-search) does provide some benefits, its effectiveness is limited. Importantly, training the entire search trajectory as a sequence poses additional challenges due to its long length, such as in the case of Countdown 5 where the search trajectories can span 60,000 tokens. Lastly, our model surpasses all previous diffusion models, demonstrating the efficacy of the multigranularity loss.

**Results on Game of 24.** As shown in Table 2, the performance of the GPT-4 with IO, CoT, and CoT-SC prompting methods from Yao et al. (2024) is unsatisfactory for the given task, with only accuracy below 10%. The introduction of ToT, which incorporates a search algorithm designed by human experts into the decoding process, significantly enhances the performance of GPT-4. This integration allows the AR model to backtrack as needed, resulting in notable improvements. However, this paradigm requires the assessment of intermediate steps using LLM, resulting in considerable computational costs due to the need for multiple LLM calls. We list the token cost in Table 2 with more details in Appendix D.1. ToT consumes 186 times more tokens than MGDM, showcasing the 'internal' search capability by promoting global consistency in diffusion modeling.

Table 2: Accuracy and token cost on game of 24.

|  | Acc. | # Token |
| --- | --- | --- |
| *Prompting* | | |
| GPT-4 IO | 7.3 | x28 |
| GPT-4 CoT | 4.0 | x61 |
| GPT-4 CoT-SC | 9.0 | x241 |
| GPT-4 ToT | 74.0 | x186 |
| *Supervised training* | | |
| GPT-2 Scratch | 18.8 | **x1** |
| **MGDM** | **76.0** | **x1** |

In summary, our model, despite having a parameter size of only 85M, significantly outperforms both the AR task-specific model of the same size (GPT-2 Scratch) in terms of performance and the larger general pre-trained model (GPT-4) in computation cost, indicating it is challenging for model scaling and decoding strategies to substitute the advantages of modeling paradigm.

## 4.2 SUDOKU

Sudoku is a classical logic-based number placement puzzle that has gained popularity due to its rigorous intellectual demands. The goal of Sudoku is to meticulously fill a $9 \times 9$ grid with numerical digits, ensuring that every column, row, and $3 \times 3$ subgrid contains all the numbers from 1 to 9.

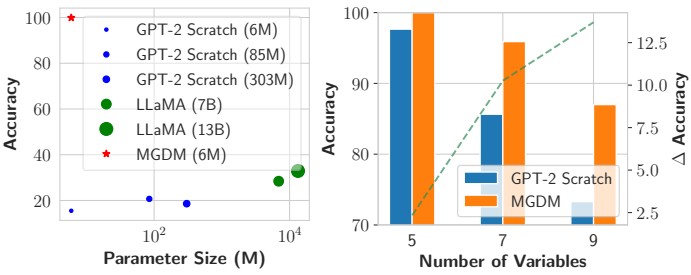

Figure 4: **(Left)** Accuracy on Sudoku. **(Right)** Accuracy on boolean satisfiability problem with increasing difficulty.

**Setup.** We collect one million solved games from Park (2016) and use the first 100k as our training set and the subsequent 1k as the testing set. We employ the digit 0 to represent the vacant position that needs to be filled. We then transform the $9 \times 9$ grid into a sequence of 81 digits, which serves as the model input. To illustrate, an example input appears as "080050060...603100007" (omitted for brevity), while the corresponding output is represented as "789251364...653184297". During tokenization, we treat each digit as a separate token.

|                                                    | Random | TopK |
| -------------------------------------------------- | ------ | ---- |
| No reweighting                                     | 82.1   | 87.3 |
| Original sequence-reweighting                      | 83.1   | 88.5 |
| + token-reweighting ($\alpha$=0.25, $\beta$=1)     | 84.9   | **90.4** |
| + token-reweighting ($\alpha$=1, $\beta$=1)        | 82.4   | 89.3 |
| + token-reweighting ($\alpha$=0.25, $\beta$=2)     | 82.4   | 87.9 |
| Linear sequence-reweighting                        | 79.6   | 87.0 |
| + token-reweighting ($\alpha$=0.25, $\beta$=1)     | 83.2   | 88.0 |
| + token-reweighting ($\alpha$=1, $\beta$=1)        | 86.7   | 90.4 |
| + token-reweighting ($\alpha$=0.25, $\beta$=2)     | 85.6   | **91.5** |

Table 3: Ablation on training reweighting strategies and inference decoding methods.

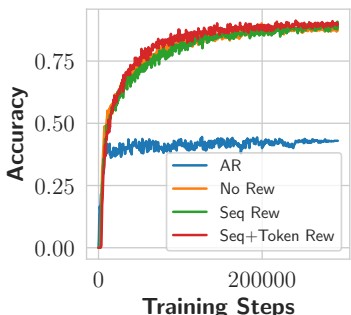

Figure 5: Evaluation accuracy throughout the training process for AR and MGDM with different reweighting strategies.

**Result.** We show the results in the left figure of Figure 4. As the size of the AR model increases, the performance remains unsatisfactory. For instance, the LLaMA model achieves a performance of only 32.9 with 13B parameters. In contrast, our model, which has only 6M parameters, is able to perfectly solve all the problems, demonstrating the significant advantage brought by the modeling architecture.

## 4.3 BOOLEAN SATISFIABILITY PROBLEM

The Boolean satisfiability problem, commonly known as SAT, is a foundational problem in computer science that has been rigorously proven to be NP-complete (Cook, 1971). This challenging combinatorial problem is attractive as a broad range of search problems from domains such as software verification, test pattern generation, planning, scheduling, and combinatorics can all routinely be solved by reducing to an appropriate SAT problem (Gomes et al., 2008). The goal of SAT is to determine whether a given Boolean formula represented in conjunctive normal form (CNF) can be assigned a set of values (0 or 1) to its variables, such that the formula evaluates to true (1). An example formula with three variables can be $(x_1 \vee \neg x_2) \wedge (\neg x_1 \vee x_2 \vee x_3) \wedge \neg x_1$ and an corresponding assignment is $x_1 = 0, x_2 = 0, x_3 = 1$.

**Setup.** Given the number of variables $n$ and clauses $m$, we iteratively sample $k = 3$ variables (and their polarities) uniformly at random until $m$ clauses are obtained. To ensure that we get relatively hard instances of SAT, we take advantage of the well-studied family of random $k$-SAT problem (Ding et al., 2015) and set the $m$ to be close to $m = 4.258n + 58.26n^{-2/3}$ given $n$, as it has been observed that SAT solvers are slow to determine the satisfiability of a formula when $m$ is near the threshold (Crawford & Auton, 1996). We consider increasing numbers of variables from $\{5,7,9\}$ and generate 50k training data for $n = 5, 7$ and 100k for $n = 9$, as well as additional 1k testing data for each $n$.

**Result.** As shown in the right figure of Figure 4, MGDM performs well in solving scenarios with five variables, while the AR model falls slightly short. As the number of variables increases, both our model and the AR model experience a certain degree of decrease in accuracy. However, the performance gap between the two models widens as the difficulty of the task increases. This indicates that our diffusion model exhibits a more pronounced advantage in handling more challenging tasks than the AR counterpart.

## 4.4 ANALYSIS

**On the Effect of Training and Decoding Strategies.** As listed in Table 3, we find that changing the sequence-reweighting strategies has only led to a slight improvement in performance. However, when a suitable parameter is selected for token-reweighting, a more significant improvement can be observed. Additionally, the easy first decoding (TopK) outperforms the random one, which aligns with previous findings (Ghazvininejad et al., 2019; Zheng et al., 2023). We compare the evaluation accuracy along the training process in Figure 5. By aligning the AR training steps with the diffu-

sion process, we can see AR converges rapidly, with the performance tends to plateau afterward. The utilization of our multi-granularity loss, which incorporates sequence and token reweighting, demonstrates superior performance, particularly during the middle stages of training. This implies that the inclusion of such a loss function contributes to enhanced convergence during the training process.

**On Decoding Speed.** We assess the trade-off between accuracy and decoding speed by comparing the performance of the AR model (GPT-2 Scratch 85M) and MGDM (85M). The speed metric is determined by the number of samples processed using a batch size of 1 on the NVIDIA GeForce V100 GPU. As shown in Figure 6(a), MGDM can flexibly control the trade-off between accuracy and decoding speed by varying the diffusion timesteps. Notably, by employing just one diffusion step, MGDM demonstrates a remarkable 10x improvement in speed compared to AR, while maintaining superior accuracy with 75% and 12.7% compared to 45.8% and 5.1% of AR on Countdown 4 and 5, respectively. We observed that the slope of countdown 4 is smaller compared to countdown 5 in the trade-off. This suggests that for tasks with lower complexity, diffusion demonstrates a more noticeable speed advantage by setting a smaller diffusion step. In addition, it also indicates that sacrificing some efficiency for performance improvements becomes particularly evident when dealing with more intricate tasks.

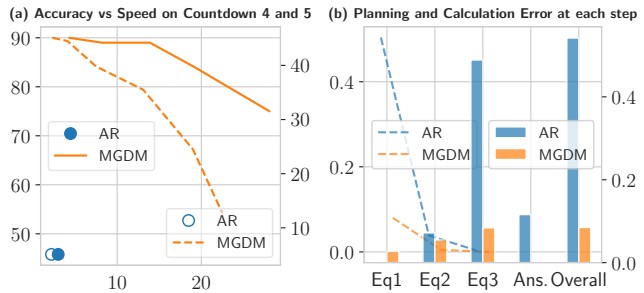

Figure 6: **(a)** Accuracy and speed (samples per second) trade-off by varying the diffusion timesteps on Countdown 4 (left y-axis) and 5 (right y-axis).**(b)** Ratio of planning (left y-axis) and calculation (right y-axis) error at each reasoning step on Countdown 4.

**Error Analysis: The Regretful Compromise.** To gain a deeper understanding of error patterns in AR and MGDM, we conducted an error analysis on Countdown 4. For instance, given the input "7,38,3,1" and the target number 14, a correct solution would be "97-38=59,59-17=42,42/3=14". We divide the solution into four parts: equation 1, equation 2, equation 3, and answer checking. First, from a calculation perspective, we assess the error ratio in each equation by comparing the left-hand side and the right-hand side, regardless of whether the correct number was chosen. As shown in Figure 6(b), the bar plot demonstrates that the majority of calculation errors for AR are concentrated in Equation 3. For example, given input "16,4,40,51" and target 87, the prediction of AR is "51-40=11,16*4=64,11+64=87" while the correct solution is "16/4=4,40+51=91,91-4=87". The first two equations are calculated correctly, but the last one is incorrectly forced to equal 87. Moreover, in many cases, the error rate for the last equation in AR (48.9%) is significantly higher than that of the previous equations (0.2% for the first and 7.2% for the second). This indicates that the model's lack of planning capability results in its realization of being unable to achieve the goal only at the end. By that point, it cannot correct the previous errors, leading it "regretfully" to rely on incorrect equations to reach the goal (e.g., the number 87 in the above case). This significantly increases the number of calculation errors in the third equation. We call this phenomenon 'The Regretful Compromise'. The reason for this is that the AR model made incorrect choices of numbers or operations in previous equations. This is demonstrated by examining the step at which the models fail the task, as depicted in Figure 6(b). It is evident that there is a notable frequency of planning errors in the first equation for the AR model, where the number of errors is significantly higher compared to our model. This highlights the limitations of the left-to-right decoding approach in AR, which adversely affects its planning ability.

## 5    RELATED WORK

### 5.1    AUTOREGRESSIVE MODELING

Starting from Bengio et al. (2000) and later Sutskever et al. (2011), the autoregressive modeling paradigm, where the prediction of a token only depends on the preceding context, is widely adopted in modeling language, until recently (OpenAI, 2022; Achiam et al., 2023; Anthropic, 2023; Team et al., 2023; Touvron et al., 2023; Jiang et al., 2023; Bai et al., 2023, *inter alia*).  Theoretically, the autoregressive Transformers have limited expressive power, but their capabilities can be expanded given sufficient chain-of-thought intermediate steps (Wei et al., 2022b; Merrill & Sabharwal, 2023; Malach, 2023).  However,  Lin et al. (2021) demonstrates that the expressing of some next-tokens requires super-polynomial computational resources and is NP-hard to approximate. Numerous advancements have been made upon the AR paradigm to compensate for modeling deficiencies, such as reverse training (Lee et al., 2023; Golovneva et al., 2024),fill-in-the-middle training (Bavarian et al., 2022), future-token prediction (Qi et al., 2020; Gloeckle et al., 2024), lookahead attention (Du et al., 2023) during the training stage, as well as search-augmented decoding (Lu et al., 2022; Xie et al., 2023; Yao et al., 2024, *inter alia*) during inference. In practice, autoregressive next-token predictors are shown to be ill-suited for planning tasks (Bubeck et al., 2023; Valmeekam et al., 2023; 2024; Dziri et al., 2024; Kambhampati et al., 2024).  Besides,  Bachmann & Nagarajan (2024); Lin et al. (2024) find not all tokens are equal and some tokens are hard to learn in the AR pretraining stage, implying the introduced subgoal imbalance phenomenon also exists in the general text corpus.

### 5.2    NON-AUTOREGRESSIVE MODELING

The non-autoregressive (NAR) generation method, which produces all target tokens simultaneously given the source context, is first proposed by  (Gu et al., 2017) in the text field for machine translation, primarily due to the efficiency consideration. While a series of advancements have been made afterward (Lee et al., 2018; Gu et al., 2019; Ghazvininejad et al., 2019; Qian et al., 2021; Huang et al., 2022, *inter alia*), traditional NAR models still tend to underperform AR models in terms of generation quality (Xiao et al., 2023).  Diffusion models (Sohl-Dickstein et al., 2015; Ho et al., 2020), a powerful class of generative models known for their impressive image-generation capabilities (Dhariwal & Nichol, 2021), have recently been applied to the field of text generation (Hoogeboom et al., 2021; Austin et al., 2021; Li et al., 2022; Campbell et al., 2022; Dieleman et al., 2022; Chen et al., 2023; Ye et al., 2023b; Lovelace et al., 2024), reinforcement learning (Janner et al., 2022; Chi et al., 2023) and protein design (Xu et al., 2022; Hoogeboom et al., 2022b; Corso et al., 2023). In essence, diffusion models perform a multi-step denoising process to progressively convert a random noise into a data sample, and the denoising procedure can be seen as parameterizing the gradients of the data distribution (Song & Ermon, 2019), connecting them to score matching (Hyvärinen & Dayan, 2005) and energy-based models (LeCun et al., 2006).  For text, the diffusion model can be seen as an extension of the traditional iterative NAR models (Gong et al., 2022) and has been shown to approach or outperform AR models on text perplexity (Han et al., 2023; Lou et al., 2023; Gulrajani & Hashimoto, 2024), diversity (Gong et al., 2022; 2023; Zhang et al., 2023) as well as various seq-to-seq tasks (Wu et al., 2023b; Zheng et al., 2023; Ye et al., 2024).  In this work, we compare diffusion with AR from a perspective of subgoal imbalance and demonstrates the effectiveness of diffusion in tasks requiring complex reasoning and planning.

## 6    CONCLUSION

This paper presents an extensive analysis of the limitations of auto-regressive (AR) language models when applied to planning tasks that involve deliberate planning, both in controlled settings and real-world contexts.  Based on an advanced understanding, we propose an improved diffusion model, named MGDM, that performs significantly better than AR and previous diffusion models on various sophisticated planning tasks. Our findings underscore the necessity to reevaluate the sequence modeling paradigm for modern large language models, especially in tackling challenging problem-solving tasks.

ACKNOWLEDGMENTS

This research was supported in part by the joint research scheme of the National Natural Science Foundation of China (NSFC) and the Research Grants Council (RGC) under grant number N_HKU714/21.

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

# A   MORE BACKGROUND AND DERIVATION OF DISCRETE DIFFUSION MODELS

## A.1   BACKGROUND

Discrete diffusion probabilistic models are first introduced in Sohl-Dickstein et al. (2015) for binary data, and later extended to categorical data in (Hoogeboom et al., 2021). Austin et al. (2021) provides a general form of discrete diffusion and introduces multiple transition matrices, including an absorbing variant that draws close connections to masked language models (Devlin et al., 2019). Several subsequent works push this line of research further from various aspects, such as incorporating editing-based operations (Johnson et al., 2022; Reid et al., 2022), casting permuted language models (Yang et al., 2019) as diffusion models (Hoogeboom et al., 2022a), developing a continuous-time framework (Campbell et al., 2022), parameterizing the routing mechanism (Zheng et al., 2023), and investigating score functions for learning the reverse process Sun et al. (2023); Lou et al. (2023).

## A.2   DERIVATION SETUP

We now provide a detailed derivation of the loss in Equation (6). For a clear illustration, we initiate derivation with a single random variable $x_0$ and ultimately link it with the multi-variable sequence $\mathbf{x}_0$. Suppose $x_0 \sim q(x_0)$ is a discrete random variable with $K$ possible categories and represented as a one-hot vector. The forward process $q(x_{1:T}|x_0) = \prod_{t=1}^{T} q(x_t|x_{t-1})$ corrupts the original data $x_0$ into a sequence of increasingly noisy latent variables $x_{1:T} := x_1, \ldots, x_T$. The learned backward process $p_\theta(x_{0:T}) = p(x_T) \prod_{t=1}^{T} p_\theta(x_{t-1}|x_t)$ gradually denoises the latent variables to the data distribution. In discrete diffusion, both the forward and backward distribution are defined as categorical distribution, e.g., $q(x_t|x_{t-1}) = \mathrm{Cat}(x_t; p = Q_t^\top x_{t-1})$ and $p_\theta(x_{t-1}|x_t) = q(x_{t-1}|x_t, f(x_t; \theta))$, where $Q_t$ is a pre-defined transition matrix of size $K \times K$ (Hoogeboom et al., 2021; Austin et al., 2021).

## A.3   THE MARGINAL AND POSTERIOR

Starting from $x_0$, we obtain the following $t$-step marginal and posterior at time $t - 1$:

$$q(x_t|x_0) = \mathrm{Cat}\left(x_t; p = \overline{Q}_t^\top x_0\right), \quad \text{with} \quad \overline{Q}_t = Q_1 Q_2 \ldots Q_t$$

$$q(x_{t-1}|x_t, x_0) = \frac{q(x_t|x_{t-1}, x_0)q(x_{t-1}|x_0)}{q(x_t|x_0)} = \mathrm{Cat}\left(x_{t-1}; p = \frac{Q_t x_t \odot \overline{Q}_{t-1}^\top x_0}{x_t^\top \overline{Q}_t^\top x_0}\right), \quad (9)$$

where $q(x_t|x_{t-1}, x_0) = q(x_t|x_{t-1})$ due to the Markov property of the forward process. The KL divergence between $q$ and $p_\theta$ can be computed by simply summing over all possible values of each random variable. The cumulative products $\overline{Q}_t$, which can be computed in closed form or precomputed for all $t$ depending on the choice $Q_t$, may be prohibitive for large $T$ and number of categories. Therefore, two commonly used forms of $Q$ are introduced by Hoogeboom et al. (2021) and Austin et al. (2021), which ensures $\overline{Q}_t$ can still be computed efficiently, allowing the framework to scale to a larger number of categories.

## A.4   TRANSITION MATRIX

Austin et al. (2021) introduced multiple types of the transition matrix $Q_t$, such as uniform (Hoogeboom et al., 2021), absorbing, discretized Gaussian and token embedding distance. The absorbing noise for discrete diffusion has been demonstrated to outperform the others (Austin et al., 2021), where the transition matrix is given by :

$$[Q_t]_{ij} = \begin{cases} 1 & \text{if} \quad i = j = m \\ 1 - \beta_t & \text{if} \quad i = j \neq m \\ \beta_t & \text{if} \quad j = m, i \neq m \end{cases}.$$

The transition matrix can also be written as $(1 - \beta_t)I + \beta_t \mathbf{1}e_m^\top$, where $e_m$ is a vector with a one on the absorbing state $m$ and zeros elsewhere. Since $m$ is an absorbing state, the corruption process

converges not to a uniform distribution but to the point-mass distribution on $m$. The transition matrices $\overline{Q} = Q_1 Q_2 \ldots Q_t$ can be computed in closed form. Specifically, we transition to another token with probability $\beta_t$ and stay the same with probability $1 - \beta_t$ in each step. After $t$ steps, the only operative quantity is the probability of not yet having transitioned to another token, given by $\alpha_t = \prod_{i=0}^{t}(1 - \beta_i)$. Therefore, we have $\overline{Q}_t = \alpha_t I + (1 - \alpha_t)\mathbf{1}e_m^\top$.

## A.5 DERIVATION OF ELBO

In order to optimize the generative model $p_{\boldsymbol{\theta}}(\boldsymbol{x}_0)$ to fit the data distribution $q(\boldsymbol{x}_0)$, we typically minimize a variational upper bound on the negative log-likelihood, defined below:

$$
\begin{aligned}
&- \log p_{\boldsymbol{\theta}}(\boldsymbol{x}_0) \\
&= - \log \int p_{\boldsymbol{\theta}}(\boldsymbol{x}_0, \boldsymbol{x}_1, \ldots, \boldsymbol{x}_T) d\boldsymbol{x}_1 \cdots d\boldsymbol{x}_T \\
&= - \log \int \frac{p_{\boldsymbol{\theta}}(\boldsymbol{x}_0, \boldsymbol{x}_1, \ldots, \boldsymbol{x}_T)}{q(\boldsymbol{x}_1, \ldots, \boldsymbol{x}_T | \boldsymbol{x}_0)} q(\boldsymbol{x}_1, \ldots, \boldsymbol{x}_T | \boldsymbol{x}_0) d\boldsymbol{x}_1 \cdots d\boldsymbol{x}_T \\
&= - \log \mathbb{E}_{q(\boldsymbol{x}_1, \ldots, \boldsymbol{x}_T | \boldsymbol{x}_0)} \left[ \frac{p_{\boldsymbol{\theta}}(\boldsymbol{x}_0, \boldsymbol{x}_1, \ldots, \boldsymbol{x}_T)}{q(\boldsymbol{x}_1, \ldots, \boldsymbol{x}_T | \boldsymbol{x}_0)} \right] \\
&\leq - \mathbb{E}_{q(\boldsymbol{x}_1, \ldots, \boldsymbol{x}_T | \boldsymbol{x}_0)} \left[ \log \frac{p_{\boldsymbol{\theta}}(\boldsymbol{x}_0, \boldsymbol{x}_1, \ldots, \boldsymbol{x}_T)}{q(\boldsymbol{x}_1, \ldots, \boldsymbol{x}_T | \boldsymbol{x}_0)} \right] \\
&= - \mathbb{E}_{q(\boldsymbol{x}_1, \ldots, \boldsymbol{x}_T | \boldsymbol{x}_0)} \left[ \log \frac{p_{\boldsymbol{\theta}}(\boldsymbol{x}_0 | \boldsymbol{x}_1) p_{\boldsymbol{\theta}}(\boldsymbol{x}_T) \prod_{t=2}^{T} p_{\boldsymbol{\theta}}(\boldsymbol{x}_{t-1} | \boldsymbol{x}_t)}{q(\boldsymbol{x}_T | \boldsymbol{x}_0) \prod_{t=2}^{T} q(\boldsymbol{x}_{t-1} | \boldsymbol{x}_t, \boldsymbol{x}_0)} \right] \\
&= - \mathbb{E}_{q(\boldsymbol{x}_1, \ldots, \boldsymbol{x}_T | \boldsymbol{x}_0)} \left[ \log p_{\boldsymbol{\theta}}(\boldsymbol{x}_0 | \boldsymbol{x}_1) - \sum_{t=2}^{T} \log \frac{q(\boldsymbol{x}_{t-1} | \boldsymbol{x}_t, \boldsymbol{x}_0)}{p_{\boldsymbol{\theta}}(\boldsymbol{x}_{t-1} | \boldsymbol{x}_t)} - \log \frac{q(\boldsymbol{x}_T | \boldsymbol{x}_0)}{p_{\boldsymbol{\theta}}(\boldsymbol{x}_T)} \right] \\
&= - \mathbb{E}_q \left[ \log p_{\boldsymbol{\theta}}(\boldsymbol{x}_0 | \boldsymbol{x}_1) - \sum_{t=2}^{T} D_{\mathrm{KL}}[q(\boldsymbol{x}_{t-1} | \boldsymbol{x}_t, \boldsymbol{x}_0) || p_{\boldsymbol{\theta}}(\boldsymbol{x}_{t-1} | \boldsymbol{x}_t)] - \underbrace{D_{\mathrm{KL}}[q(\boldsymbol{x}_T | \boldsymbol{x}_0) || p(\boldsymbol{x}_T)]}_{L_T \text{(const)}} \right] \\
&= - \underbrace{\mathbb{E}_{q(\boldsymbol{x}_1 | \boldsymbol{x}_0)} \log p_{\boldsymbol{\theta}}(\boldsymbol{x}_0 | \boldsymbol{x}_1)}_{L_0} + \sum_{t=2}^{T} \underbrace{\mathbb{E}_{q(\boldsymbol{x}_t | \boldsymbol{x}_0)} \left[ D_{\mathrm{KL}}[q(\boldsymbol{x}_{t-1} | \boldsymbol{x}_t, \boldsymbol{x}_0) || p_{\boldsymbol{\theta}}(\boldsymbol{x}_{t-1} | \boldsymbol{x}_t)] \right]}_{L_{t-1}} + L_T \text{(const)}.
\end{aligned}
$$

## A.6 DERIVATION FOR EQUATION (6)

The categorical distribution $q(\boldsymbol{x}_{t-1} | \boldsymbol{x}_t, \boldsymbol{x}_0)$ based on Equation (9) is given as:

$$
\begin{aligned}
&q(\boldsymbol{x}_{t-1} | \boldsymbol{x}_t, \boldsymbol{x}_0) \\
&= \frac{Q_t \boldsymbol{x}_t \odot \overline{Q}_{t-1}^\top \boldsymbol{x}_0}{\boldsymbol{x}_t^\top \overline{Q}_t^\top \boldsymbol{x}_0} \\
&= \frac{[(1 - \beta_t)\boldsymbol{x}_t + \beta_t \sigma_{\boldsymbol{x}_t} \mathbf{1}] \odot [\alpha_{t-1}\boldsymbol{x}_0 + (1 - \alpha_{t-1})e_m]}{\alpha_t \boldsymbol{x}_t^\top \boldsymbol{x}_0 + (1 - \alpha_t)\boldsymbol{x}_t^\top e_m} \\
&= \frac{(1 - \beta_t)\alpha_{t-1}\boldsymbol{x}_t \odot \boldsymbol{x}_0 + (1 - \beta_t)(1 - \alpha_{t-1})\boldsymbol{x}_t \odot e_m + \beta_t \alpha_{t-1}\sigma_{\boldsymbol{x}_t}\mathbf{1} \odot \boldsymbol{x}_0 + \beta_t(1 - \alpha_{t-1})\sigma_{\boldsymbol{x}_t}\mathbf{1} \odot e_m}{\alpha_t \boldsymbol{x}_t^\top \boldsymbol{x}_0 + (1 - \alpha_t)\boldsymbol{x}_t^\top e_m} \\
&= \frac{(1 - \beta_t)\alpha_{t-1}\boldsymbol{x}_t \odot \boldsymbol{x}_0 + (1 - \beta_t)(1 - \alpha_{t-1})\sigma_{\boldsymbol{x}_t}\boldsymbol{x}_t + \beta_t \alpha_{t-1}\sigma_{\boldsymbol{x}_t}\boldsymbol{x}_0 + \beta_t(1 - \alpha_{t-1})\sigma_{\boldsymbol{x}_t}e_m}{\alpha_t \boldsymbol{x}_t^\top \boldsymbol{x}_0 + (1 - \alpha_t)\sigma_{\boldsymbol{x}_t}},
\end{aligned}
$$

where $\sigma_{\boldsymbol{x}_t} := e_m(\boldsymbol{u} = \boldsymbol{x}_t)$ represents the probability of noise drawn from $e_m$ being equal to $\boldsymbol{x}_t$. Note $\boldsymbol{x}_t \odot \boldsymbol{x}_0 = 0$ if $\boldsymbol{x}_t \neq \boldsymbol{x}_0$ otherwise 1. Thus the computation of $q(\boldsymbol{x}_{t-1} | \boldsymbol{x}_t, \boldsymbol{x}_0)$ breaks down into two cases:

$$
q(\boldsymbol{x}_{t-1} | \boldsymbol{x}_t, \boldsymbol{x}_0) = \begin{cases} \eta_t \boldsymbol{x}_t + (1 - \eta_t) e_m, & \text{if } \boldsymbol{x}_t = \boldsymbol{x}_0 \\ \lambda_t \boldsymbol{x}_0 + (1 - \lambda_t) e_m(\boldsymbol{x}_t), & \text{if } \boldsymbol{x}_t \neq \boldsymbol{x}_0, \end{cases}
$$

where $\eta_t := 1 - \frac{\beta_t(1-\alpha_{t-1})e_m(\boldsymbol{u}=\boldsymbol{x}_t)}{\alpha_t+(1-\alpha_t)e_m(\boldsymbol{u}=\boldsymbol{x}_t)}$, $\lambda_t := \frac{\alpha_{t-1}-\alpha_t}{1-\alpha_t}$, and $e_m(\boldsymbol{x}_t) = (1-\beta_t)\boldsymbol{x}_t + \beta_t e_m$ denotes a noise distribution that interpolates between $\boldsymbol{x}_t$ and $e_m$.

Recall the distribution $p_{\boldsymbol{\theta}}(\boldsymbol{x}_{t-1}|\boldsymbol{x}_t)$ is parameterized by $q(\boldsymbol{x}_{t-1}|\boldsymbol{x}_t, f(\boldsymbol{x}_t;\boldsymbol{\theta}))$, the KL divergence between $q(\boldsymbol{x}_{t-1}|\boldsymbol{x}_t,\boldsymbol{x}_0)$ and $p_{\boldsymbol{\theta}}(\boldsymbol{x}_{t-1}|\boldsymbol{x}_t)$ becomes 0 when $\boldsymbol{x}_t = \boldsymbol{x}_0$. In the case of absorbing diffusion, $\boldsymbol{x}_t = e_m$ if $\boldsymbol{x}_t \neq \boldsymbol{x}_0$ and $e_m(\boldsymbol{x}_t) = e_m$. $q(\boldsymbol{x}_{t-1}|\boldsymbol{x}_t,\boldsymbol{x}_0)$ has probability $\lambda_t$ on index $x_0$ and $1-\lambda_t$ on the absorbing state. The model $f(\boldsymbol{x}_t;\boldsymbol{\theta})$ has zero-probability on the absorbing state as it never predicts the mask token. Therefore, $p_{\boldsymbol{\theta}}(\boldsymbol{x}_{t-1}|\boldsymbol{x}_t)$ also has $1-\lambda_t$ probability on the absorbing state. Putting them together, we derive the KL divergence as:

$$D_{\mathrm{KL}}[q(\boldsymbol{x}_{t-1}|\boldsymbol{x}_t,\boldsymbol{x}_0)||p_{\boldsymbol{\theta}}(\boldsymbol{x}_{t-1}|\boldsymbol{x}_t)] = 1_{x_t \neq x_0}[\lambda_t \log \frac{\lambda_t}{f(\boldsymbol{x}_t;\boldsymbol{\theta})_{x_0}} + (1-\lambda_t)\log\frac{1-\lambda_t}{1-\lambda_t}]$$
$$= -\lambda_t 1_{x_t \neq x_0}\boldsymbol{x}_0^\top \log f(\boldsymbol{x}_t;\boldsymbol{\theta}) + C,$$

where $1_{x_t \neq x_0}$ is 1 if $x_t \neq x_0$ otherwise 0, and $C$ is a constant. Moreover, given $\alpha_0 = 1$ by definition and $\lambda_0 = 1$, we have:

$$L(\boldsymbol{x}_0) = -\mathbb{E}_{q(\boldsymbol{x}_0)} \sum_{t=1}^T \lambda_t \mathbb{E}_{q(\boldsymbol{x}_t|\boldsymbol{x}_0)} 1_{\boldsymbol{x}_t \neq \boldsymbol{x}_0} \boldsymbol{x}_0^\top \log f(\boldsymbol{x}_t;\boldsymbol{\theta})$$

for a single random variable, and

$$L(\mathbf{x}_0) = -\sum_{n=1}^N \mathbb{E}_{q(\mathbf{x}_{0,n})} \sum_{t=1}^T \lambda_t \mathbb{E}_{q(\mathbf{x}_{t,n}|\mathbf{x}_{0,n})} 1_{\mathbf{x}_{t,n} \neq \mathbf{x}_{0,n}} \mathbf{x}_{0,n}^\top \log f(\mathbf{x}_{t,n};\boldsymbol{\theta})$$

for $\mathbf{x}_0$ that represents a sequence of random variables $\mathbf{x}_0 = (\boldsymbol{x}_{0,1}, \ldots, \boldsymbol{x}_{0,N})$, where the $\lambda_t$ also represents the reweighting term $w(t)$ in Equation (6).

## B   ALGORITHMS FOR TRAINING AND INFERENCE

The detailed algorithms for training and inference are illustrated in Algorithm 1 and 2, respectively. For conditional training and inference, we split $\mathbf{x}$ into $[\mathbf{x}^{\mathrm{src}}; \mathbf{x}^{\mathrm{tgt}}]$ and freeze the condition part $\mathbf{x}^{\mathrm{src}}$ during training and inference.

---

**Algorithm 1** Training MGDM

**Input:** neural network $f(\cdot;\boldsymbol{\theta})$, data distribution $p_{\mathrm{data}}(\mathbf{x}_{0,1:N})$, a custom sequence reweighting term $w(t)$, token reweighting parameters $\alpha$ and $\gamma$, timesteps $T$.
**Output:** model parameters $\boldsymbol{\theta}$.
**repeat**
    Draw $\mathbf{x}_{0,1:N} \sim p_{\mathrm{data}}(\mathbf{x}_{0,1:N})$;
    Draw $t \in \mathrm{Uniform}(\{1,\ldots,T\})$;
    Draw $\mathbf{x}_t \sim q(\mathbf{x}_t|\mathbf{x}_0)$;
    **for** $n = 1,2,\ldots,N$ **do**
        Let $u(\mathbf{x}_0,\mathbf{x}_t,n;\boldsymbol{\theta}) := \mathbf{1}_{\mathbf{x}_{t,n} \neq \mathbf{x}_{0,n}} \mathbf{x}_{0,n}^\top \log f(\mathbf{x}_t;\boldsymbol{\theta})_n$;
        Let $v(\mathbf{x}_{t,n}) = \alpha(1 - \exp u(\mathbf{x}_0,\mathbf{x}_t,n;\boldsymbol{\theta}))^\gamma$;
    **end for**
    $L_{\boldsymbol{\theta}} = -w(t) \sum_{n=1}^N v(\mathbf{x}_{t,n}) u(\mathbf{x}_0,\mathbf{x}_t,n;\boldsymbol{\theta})$;
    Minimize $L_{\boldsymbol{\theta}}$ with respect to $\boldsymbol{\theta}$;
**until** converged

---

---

**Algorithm 2** Sampling from MGDM

---

**Input:** trained network $f(\cdot;\boldsymbol{\theta})$, mask token id $\boldsymbol{m}$, timesteps $T$, temperature $\tau$.
**Output:** generated sample $\mathbf{x}_0$.
**for** $n = 1, 2, \ldots, N$ **do**
    Initialize $\mathbf{x}_{T,n} = \boldsymbol{m}$;
**end for**
**for** $t = T, \ldots, 1$ **do**
    Define indicator $\mathbf{e}_t = \mathrm{TopK}\left(f(\mathbf{x}_t;\boldsymbol{\theta})\right)$ with indices in top-$t/T$ values set to 1 and others 0;
    **for** $n = 1, 2, \ldots, N$ **do**
        Draw $\widetilde{\mathbf{x}}_{0,n} \sim \mathrm{Categorical}\left(f(\mathbf{x}_t;\boldsymbol{\theta})/\tau\right)$;
        $\mathbf{x}_{t-1,n} = \mathbf{e}_{t,n}\widetilde{\mathbf{x}}_{0,n} + (1 - \mathbf{e}_{t,n})\boldsymbol{m}$;
    **end for**
**end for**
**Return** $\mathbf{x}_{0,1:N}$.

---

## C  ADDITIONAL EXPERIMENTAL DETAILS

### C.1  TASK DETAILS

Table 4: Dataset statistics. Minimal and CD are short for the minimal planning task and Countdown, respectively.

|                  | Minimal | CD3  | CD4  | CD5  | Sudoku | 3-SAT 5v | 3-SAT 7v | 3-SAT 9v |
|------------------|---------|------|------|------|--------|----------|----------|----------|
| Train Instance   | 50k     | 500k | 500k | 500k | 100k   | 50k      | 50k      | 100k     |
| Test Instance    | 1k      | 1k   | 1k   | 1k   | 1k     | 1k       | 1k       | 1k       |
| Avg Input Token  | 47      | 11   | 13   | 16   | 81     | 245      | 269      | 305      |
| Avg Output Token | 21      | 16   | 25   | 35   | 81     | 9        | 13       | 17       |
| Max Input Token  | 49      | 12   | 15   | 18   | 81     | 245      | 269      | 305      |
| Max Output Token | 23      | 22   | 35   | 52   | 81     | 9        | 13       | 17       |

We show the statistics and input-output examples on each dataset in Table 4 and Table 10, respectively.

Table 5: Model parameters with varying model size.

|               | Tiny | Base | Medium |
|---------------|------|------|--------|
| Parameters    | 6M   | 85M  | 303M   |
| Num of Layer  | 3    | 12   | 24     |
| Num of Head   | 12   | 12   | 16     |
| Hidden Dim    | 384  | 768  | 1024   |

### C.2  MGDM IMPLEMENTATION DETAILS

We conduct all the experiments on NVIDIA V100-32G GPUs, and we use 8 GPUs for training and sampling. We mainly consider comparing diffusion and AR models trained from scratch with different model sizes, with arguments for each size listed in Table 5. We use the GPT-2 architecture for both MGDM and AR. We set the learning rate to 1e-3 for the tiny model and 3e-4 for others, and we set the batch size to 1024 across all the models and tasks. We train MGDM for 1200 epochs on the minimal planning task, 300 epochs on Sudoku, and 600 epochs on other datasets. By default, we set the diffusion sampling steps to $T = 20$ for tasks with average output tokens larger than 20, otherwise $T = 10$. We use a decoding temperature $\tau = 0.5$ for all tasks. For all the experiments, we have verified the statistical significance by running them multiple times.

### C.3 BASELINE IMPLEMENTATION DETAILS

We train the AR model until convergence, and the number of training epochs is set to 200 for the minimal planning task, 300 for SAT, and 40 for others. We keep other parameters, e.g., batch size and learning rate, the same as training MGDM.

For LLaMA (Touvron et al., 2023), we use LoRA fine-tuning (Hu et al., 2021) with lora rank setting to 16. We use a learning rate of 1e-4, a batch size of 256, and train for a maximum of 20 epochs to ensure the model has converged. For GPT-4, we borrow the numbers from Yao et al. (2024).

For all the diffusion baselines, we use the same transformer architecture as GPT-2 to control the variables. We set the training parameters the same as MGDM, e.g., number of training epochs to 600, learning rate to 3e-4, and batch size to 1024. During inference, we set decoding timesteps to 20 for all diffusion models as we didn't observe a clear performance improvement when scaling to 1024.

## D ADDITIONAL EXPERIMENTS

### D.1 TOKEN CONSUMPTION ON GAME OF 24

We show the detailed accuracy and token consumption on the game of 24 in Table 6.

Table 6: Detailed accuracy and token consumption on game of 24.

|  | Accuracy | Prompt Tokens | Generate Tokens |
|---|---|---|---|
| GPT-4 IO | 7.3 | 1k | 18 |
| GPT-4 CoT | 4.0 | 2.2k | 67 |
| GPT-4 CoT-SC | 9.0 | 2.2k | 6.7k |
| GPT-4 ToT | 74.0 | 1.4k | 2.5k |
| GPT2-Scratch | 18.8 | 11 | 26 |
| MGDM | 76.0 | 11 | 26 |

### D.2 AR WITH TOKEN REWEIGHTING

We show the accuracy of AR with the same token reweighting mechanism in Equation 8 on the minimal planning task in Table 7. We find that applying token reweighting to AR models still cannot solve subgoals with PD larger than 1 (i.e., with accuracy around 50%), similar to the original AR.

Table 7: Results of AR with token reweighting.

| Planning Distance | AR | AR with token reweighting |
|---|---|---|
| 0 | 100 | 100 |
| 1 | 100 | 100 |
| 2 | 51.1 | 52.1 |
| 3 | 46.9 | 51.5 |
| 4 | 52.0 | 50.3 |
| 5 | 49.9 | 51.9 |

## D.3 SCALING BOTH DATA AND MODEL SIZE

As an extension of Table 1, we show the accuracy of AR and MGDM when both data and model size are increased in Table 8. We find scaling both data and model size is effective for both AR and MGDM.

Table 8: Results of scaling both model and data size.

|  | AR | MGDM |
|---|---|---|
| 85M model, 500k data | 45.8 | 91.5 |
| 303M model, 500k data | 41.3 | 88.3 |
| 303M model, 1M data | 53.3 | 95.6 |

## D.4 CASE STUDY

Table 9: Example predictions on Countdown 4. For each sub-equation, we mark the planning error in red and the calculation error in **bold**. AR exhibits more calculation errors in the last equation due to incorrect planning in the previous steps.

| Numbers | Goal | Groundtruth | AR Prediction | MGDM Prediction |
|---|---|---|---|---|
| 64,36,52,42 | 14 | 64-52=12,36/12=3,42/3=14 | **64/36=2**,52/2=26,**42-26=14** | 64-52=12,36/12=3,42/3=14 |
| 9,73,99,75 | 81 | 75-73=2,9*2=18,99-18=81 | 99+75=174,**174/9=16**,73+16=81 | 75-73=2,9*2=18,99-18=81 |
| 2,52,20,73 | 57 | 52-20=32,32/2=16,73-16=57 | 2*20=40,73-52=21,**40+21=57** | 52-20=32,32/2=16,73-16=57 |
| 9,80,4,5 | 89 | 9+80=89,5-4=1,89*1=89 | 9-5=4,4/4=1,80+1=81 | 9+80=89,5-4=1,89*1=89 |
| 65,2,61,22 | 96 | 65-61=4,2+22=24,4*24=96 | 65-61=4,22*4=88,2+88=90 | 65-61=4,2+22=24,4*24=96 |
| 42,47,9,14 | 81 | 47-42=5,14-5=9,9*9=81 | 47-42=5,14*5=70,**9+70=89** | 47-42=5,14*5=70,**9+70=81** |
| 41,4,48,20 | 96 | 41*4=164,48+20=68,164-68=96 | 48-41=7,20-7=13,**4*13=92** | 4*20=80,**41-40=2**,48*2=96 |
| 21,36,3,42 | 39 | 42-36=6,3*6=18,21+18=39 | 36-21=15,15/3=5,42-5=37 | 42-21=21,36/3=12,**21-12=39** |

We show more prediction cases of the autoregressive model and our model in Table 9.

Table 10: Task details by showing example input and output for each dataset.

| Task | Input Example | Output Example |
|---|---|---|
| Minimal Planning | 2,10/10,4/11,5/2,0/8,2/0,11/6,2/1,9/5,3/4,1-8,3 | 8,2/2,0/0,11/11,5/5,3 |
| Countdown 3 | 15,44,79,50 | 44-15=29,79-29=50 |
| Countdown 4 | 86,28,13,31,96 | 86+28=114,31-13=18,114-18=96 |
| Countdown 5 | 50,36,82,44,31,51 | 44-36=8,82*31=2542, 8+2542=2550,2550/50=51 |
| Sudoku | 080050060
460907108
005000029
970006500
000872031
300049000
004025003
010000480
603100007 | 789251364
462937158
135468729
978316542
546872931
321549876
894725613
217693485
653184297 |
| 3-SAT 5v | 1,4,5/1,-4,-5/2,-4,5/-1,-2,5/3,4,5/-2,-4,-5/2,3,-4/-2,-3,5/1,2,4/1,-2,3/-1,3,5/1,-2,-4/1,4,-5/1,-2,-5/1,2,-5/-1,-3,-4/-1,3,-5/-1,3,4/2,-4,-5/-1,-4,5/1,-3,-5/1,3,-5/1,-3,-4/-2,3,5/1,2,5/-1,2,-4/1,-2,4/1,-4,5/3,4,-5/-1,2,-3/1,-3,5/-2,4,5/1,-2,5/-1,2,5/1,3,-4/-1,-4,-5/-2,-3,-4/2,4,5/-2,3,-4/-3,4,5/2,-3,5 | 1,2,3,-4,5 |
| 3-SAT 7v | -2,-3,-7/2,-4,-7/-3,4,-5/1,2,-3/1,5,-7/-5,-6,-7/2,-5,6/2,-5,-6/-3,-4,6/-1,2,-4/-3,6,7/-2,-5,6/2,3,-7/-1,2,3/-2,3,-4/-1,3,7/1,-2,-7/2,4,6/1,2,-7/2,-3,-6/1,-2,6/-1,5,7/3,-6,-7/2,6,7/-2,-6,-7/-2,3,-5/3,5,-6/-2,6,-7/-1,-2,-7/5,-6,-7/2,-6,-7/-2,5,7/-3,-4,5/2,3,-4/-3,5,-7/3,-4,5/-2,3,-6/1,2,-6/1,4,-7/1,4,7/2,4,5/1,5,-6/1,3,4/2,3,7/1,-2,4 | 1,2,3,4,5,6,-7 |
| 3-SAT 9v | 3,-4,-6/1,3,5/2,-7,8/1,-3,6/2,-3,-8/-4,-5,-7/1,-6,-9/1,8,-9/2,3,-9/3,-5,9/-3,7,9/-2,-3,9/-1,-5,-9/-2,-7,-9/-1,3,5/2,-5,-9/4,-7,-9/-2,3,-8/2,3,7/-4,6/-2,3,5/-2,-6,-8/-3,-4,-8/-2,6,7/-3,4,6/-3,-6,9/2,7,-9/2,4,-5/-3,-5,8/-4,5,-7/-4,-6,-8/2,-6,9/2,-5,9/1,4,-9/5,8,9/1,-6,7/-3,6,-9/1,4,-5/4,-6,9/-1,2,6/1,-2,-5/1,-2,-9/-4,7,9/-1,-4,-7/-3,5,-8/-1,-3,6/-2,-3,6/-3,6,9/-1,-5,8/1,-5,-9/1,4,8 | 1,2,3,4,-5,6,-7,-8,9 |

