# OpenReview forum: "Beyond Autoregression: Discrete Diffusion for Complex Reasoning and Planning"
_ICLR.cc/2025/Conference — ICLR 2025 Poster_

### Official Review · Reviewer_gxqG · 2024-10-26

**Soundness:** 2
**Presentation:** 3
**Contribution:** 3
**Rating:** 6
**Confidence:** 3

**Summary:**

The authors begin by pointing out that autoregressive (AR) models struggle with reasoning and long-term planning tasks. The authors propose _discrete diffusion (DD)_ as a better modeling approach for these classes of tasks.

They begin their contributions by providing a theoretical argument for their assertion from the viewpoint of what they deem "subgoal imbalance". Here, they argue that the subgoals faced when attempting reasoning and planning tasks autoregressively are not all necessarily of equal learning difficulty. The authors follow this initial theoretical setup with empirical evidence on a toy synthetic planning task demonstrating that autoregressive models struggle when the subgoals become more difficult, e.g. require longer planning horizons.

The authors pose the same toy task to a DD approach, which they demonstrate does not show similar signs of failure as the AR approach. Given this, they continue their contributions by theoretically outlining _why_ the DD approach does not struggle as much as the AR model on the same toy task. Here, they compare the losses of the AR approach and DD approach, and point out that the DD loss remains low compared to its AR counterpart. Moreover, they show that the unweighted loss of the DD approach reveals the following trend: as the number of timesteps increases, subgoals at smaller timesteps become easier to learn (face lower loss). The authors analyze this from a "multi-view learning" perspective, arguing that each x_t in DD can be interpreted as a different view of x_0. This, they argue, affords more effective learning of challenging subgoals.

Given these insights, the authors develop an extension to discrete diffusion tailored to planning and reasoning where the "subgoal imbalance" problem supposedly emerges. Their extension, dubbed multi-granularity diffusion modeling (MDM) consists in introducing a token-level reweighting mechanism in the Monte-Carlo sampled DD loss equation, designed to adaptively reduce the loss for easy tokens while emphasizing the loss for harder ones.

The authors then test their method on 3 "real-world scenarios", namely on Countdown, Sudoku and SAT. They compare MDM to existing diffusion models as well as a number of AR baselines. Throughout, the authors empirically demonstrate the superiority of DD approaches to these planning and reasoning tasks, as well as the performance edge gained with MDM over existing DD approaches.

**Strengths:**

- The paper is very well structured. Ideas are presented in a organised, sequential manner with little to no jumping required. Evidence and explanations are provided soon after assertions. The positioning of every section makes sense.
-  The paper is original in some ways: it contributes this novel concept of "subgoal imbalance" and later links "multi-view learning" as a possible explanation for diffusion's superiority over autoregressive methods when it comes to planning and reasoning tasks.
- The main contribution of the paper: theoretical argumentation supported by empirical evidence for why diffusion works better than autoregression for planning and reasoning is a significant contribution considering the value of such capabilities, and may provide a further shift towards diffusion methods in the field.

**Weaknesses:**

- I found that the definition of "subgoal imbalance" was a bit awkward -- is this something that can only be talked about from an autoregressive perspective or is this a general aspect of multi-step tasks such as planning and reasoning? The more formal proposition at line 160 makes it sound like the former, but throughout the text my understanding leans more to the latter. I recommend trying to make this proposition more rigorous, more carefully outlining when it applies and when it does not.
- I found the "multi-view learning" perspective a bit underdeveloped. I think it would help to define what "view" means in this context, rather than expecting the reader to refer to Xu et al. 2013. Especially given how well the rest of the paper flows. In general I think this part of the paper (L243-252) could have gone into more depth for readers to get a better understanding of what the authors were intending, because at the moment it reads as a bit vague and non-rigorous.
- Perhaps because I did not fully grasp the "multi-view learning" perspective due to its underdevelopment, it is not 100% clear to me how what is observed in 3.2 led to the insights on which MDM is developed in 3.3. Perhaps it would have been better to drop the "multi-view learning" perspective and focus entirely on the subgoal imbalance idea, which I think transitions naturally to the extension presented in 3.3 without much of the argumentation in 3.2 needed.
- I am confused by the inconsistency of the baselines in section 4. It would have been nice to always use all the baselines, but for some reason e.g. GPT-4 ToT is only used on Game of 24.
- Similarly, since there are no claims of the generality of diffusion models which is on the other hand often touted as an emergent property of the autoregressive models, it would have been nice to have other non-general, planning-specific baselines to compare against. E.g. the question I am getting at is, many "real-world" scenarios could be solved analytically a Planning Domain Definition Language (PDDL), so why should one use MDM (or other diffusion methods) instead?

**Questions:**

- I've already hinted this a little bit above, but one of the properties of these very popular AR models is their emergent generality: they can be successfully applied to a variety of downstream tasks while being trained on a simple autoregressive objective. In the paper you have shown that _when training specifically for planning_ discrete diffusion is a better approach than autoregression. My question is -- should we expect this finding to generalize to "general" diffusion solutions too? I.e. suppose one has some foundation *diffusion* LM -- do we expect such an LM to still be better at planning than an equivalent AR LM? Or is the assertion/result of the paper only for the narrow case of planning specific training?
- If indeed this paper is about the narrow case, in which way, if any, would you envision incorporating this narrow case into a more general model capable of more than just planning?

---

> ### Author Response · Authors · 2024-11-22
>
> We sincerely thank Reviewer gxqG for your detailed review and are grateful for the time you spent on our submission. We are also glad to know you think our paper is very well structured and original. Below we would like to give detailed responses to each of your comments.
>
> **Weaknesses1: I found that the definition of "subgoal imbalance" was a bit awkward -- is this something that can only be talked about from an autoregressive perspective or is this a general aspect of multi-step tasks such as planning and reasoning? The more formal proposition at line 160 makes it sound like the former, but throughout the text my understanding leans more to the latter. I recommend trying to make this proposition more rigorous, more carefully outlining when it applies and when it does not.**
>
> Thanks for your great suggestion. The subgoal imbalance is a general aspect of multi-step tasks like planning and reasoning, where different subgoals vary in difficulty, as shown in Figure 2. This imbalance exists regardless of whether the tasks are modeled autoregressively or using other approaches—it is inherent to the data and the nature of the tasks themselves.
> Since our work focuses on the weaknesses of the autoregressive (AR) paradigm, we explained subgoal imbalance from this perspective.
>
> We intend to illustrate how subgoal imbalance particularly affects AR models due to their sequential processing nature. We acknowledge that the formal proposition at line 160 may give the impression that subgoal imbalance is specific to AR models. To address this, we have revised the proposition in the updated manuscript to involve only the data distribution, emphasizing that subgoal imbalance is a property of data distribution, not tied to any specific modeling approach.
>
>
> **Weaknesses2: I found the "multi-view learning" perspective a bit underdeveloped. I think it would help to define what "view" means in this context, rather than expecting the reader to refer to Xu et al. 2013. Especially given how well the rest of the paper flows. In general I think this part of the paper (L243-252) could have gone into more depth for readers to get a better understanding of what the authors were intending, because at the moment it reads as a bit vague and non-rigorous.**
>
> Thank you for your suggestion. We have added detailed descriptions in the updated manuscript. Each $x_t$ can be interpreted as a distinct view of $x_0$, where each view provides different information about $x_0$. In the diffusion process, by exploring the consistency and complementary properties of different views offered by a diverse range of interrelated objectives $u(x_0,x_t, n;\theta)$, our findings suggest that objectives challenging to learn in AR models become more feasible to learn in diffusion models.
>
> **Weaknesses3: Perhaps because I did not fully grasp the "multi-view learning" perspective due to its underdevelopment, it is not 100% clear to me how what is observed in 3.2 led to the insights on which MDM is developed in 3.3. Perhaps it would have been better to drop the "multi-view learning" perspective and focus entirely on the subgoal imbalance idea, which I think transitions naturally to the extension presented in 3.3 without much of the argumentation in 3.2 needed.**
>
> Thank you for your suggestion. The multi-view learning perspective is an advantage introduced by diffusion models, as they naturally capture multiple representations (views) of the data. Our MDM method builds on this by applying weighting to the more challenging views to effectively address subgoal imbalance. We have clarified this connection in the revised manuscript to better explain how the observations in Section 3.2 led to the development of MDM in Section 3.3.
>
> **Weaknesses4 I am confused by the inconsistency of the baselines in section 4. It would have been nice to always use all the baselines, but for some reason e.g. GPT-4 ToT is only used on Game of 24.**
>
> Thanks for your feedback. Our main objective was to analyze the performance of autoregressive (AR) models versus diffusion models under fair and comparable conditions, focusing on models of the same scale to ensure a balanced and affordable comparison. GPT-4 with Tree of Thought (ToT) was included only for the Game of 24 as a bonus reference to demonstrate that even larger AR models rely heavily on search strategies to perform well on complex tasks. This highlights that without such search mechanisms, large AR models may struggle, whereas our diffusion-based approach achieves strong performance without additional search.

---

> > ### Author Response · Authors · 2024-11-22
> >
> > **Weaknesses5: Similarly, since there are no claims of the generality of diffusion models which is on the other hand often touted as an emergent property of the autoregressive models, it would have been nice to have other non-general, planning-specific baselines to compare against. E.g. the question I am getting at is, many "real-world" scenarios could be solved analytically a Planning Domain Definition Language (PDDL), so why should one use MDM (or other diffusion methods) instead?**
> >
> > Thanks for your insightful comment. We acknowledge that certain real-world scenarios can be solved by translating to formal language like PDDL. However, these formal methods are not comprehensive in some way and may not be suitable for all types of planning tasks. For instance, while Sudoku can be solved through a simple DFS search algorithm, expressing more general text planning problems in PDDL can be challenging due to the complexity and diversity of real-world tasks.
> >
> > Our aim with MDM is to enhance the model's intrinsic planning abilities without relying solely on formal representations or domain-specific solutions. However, we acknowledge the importance of generalizability and will use the insights from this paper as a foundation to investigate scaling and generality in our future research.
> >
> > **Questions1:I've already hinted this a little bit above, but one of the properties of these very popular AR models is their emergent generality: they can be successfully applied to a variety of downstream tasks while being trained on a simple autoregressive objective. In the paper you have shown that when training specifically for planning discrete diffusion is a better approach than autoregression. My question is -- should we expect this finding to generalize to "general" diffusion solutions too? I.e. suppose one has some foundation diffusion LM -- do we expect such an LM to still be better at planning than an equivalent AR LM? Or is the assertion/result of the paper only for the narrow case of planning specific training?**
> >
> > Thanks for your great question. Our ultimate goal is to develop diffusion LLMs that outperform equivalent AR LLMs in both general problem-solving and complex planning tasks, not just in the narrow case of task-specific training. Due to computational resource constraints, we began by investigating the potential of this modeling approach in complex reasoning tasks from a fundamental way, e.g., the insights on why planning is hard for AR, the design of minimal planning task, and the explanation of how diffusion works both theoretically and empirically. We believe that with further scaling on a large general corpus, diffusion LMs could exhibit superior planning capabilities compared to AR LLMs across a broader range of applications.
> >
> >
> > **Questions2: If indeed this paper is about the narrow case, in which way, if any, would you envision incorporating this narrow case into a more general model capable of more than just planning?**
> >
> > Yes, we envision integrating our approach into a more general model capable of more than just planning. By increasing the model size and incorporating general-purpose training data, we aim to enhance its abilities across a broader range of tasks while maintaining the planning advantages demonstrated in our current work. We are actively working on this expansion and believe that scaling up the model and training on diverse corpora will enable it to perform effectively on general tasks beyond planning.

---

> > > ### Author Response · Authors · 2024-11-25
> > >
> > > Dear Reviewer,
> > >
> > > We would like to first express our sincere gratitude for your time and effort in reviewing our paper, and we truly appreciate your constructive feedback regarding our work.
> > >
> > > As the author-reviewer discussion period is coming to a close, we wonder if you could kindly share some of your thoughts so we can keep the discussion rolling to address your concerns if there are any. We are eager to address any additional questions or issues you may have.
> > >
> > > We would greatly appreciate it if you would consider adjusting the score based on our responses and the other review comments.
> > >
> > > Thank you once again for your constructive feedback!
> > >
> > > Sincerely,
> > >
> > > The Authors

---

> > > > ### Author Response · Authors · 2024-12-02
> > > > **Looking forward to further discussion**
> > > >
> > > > Dear Reveiwer gxqG,
> > > >
> > > > We would like to thank you again for your detailed reviews. Since the rebuttal deadline is approaching soon (Dec 2 AoE), we would appreciate it if you could kindly share some of your thoughts so we can keep the discussion rolling to address your concerns if there are any.
> > > >
> > > > For your reference, we make a summary of our paper's contribution:
> > > > - The insights on why planning is hard for AR through the lens of subgoal imbalance
> > > > - The design of minimal planning task to validate the shortcomings of AR
> > > > - The explanation of how diffusion learns subgoals that challenge the AR model both theoretically and empirically
> > > > - The proposing of Multi-granularity Diffusion Modeling (MDM), and showing it significantly outperforms autoregressive models on more complex reasoning challenges
> > > >
> > > > As most of the current generation of LLMs are AR-based, our work highlights the potential of diffusion-based approaches in advancing AI capabilities for sophisticated language understanding and problem-solving tasks, and we hope to promote the development of the next generation of diffusion-based LLMs.
> > > >
> > > > Thanks a lot for your time! Looking forward to your reply.
> > > >
> > > > Best regards,
> > > >
> > > > Authors

---

### Official Review · Reviewer_aoA3 · 2024-11-03

**Soundness:** 4
**Presentation:** 4
**Contribution:** 4
**Rating:** 8
**Confidence:** 3

**Summary:**

This work proposes to use a form of diffusion models as an alternative to auto-regressive models for reasoning and planning problems. The authors illustrate the weaknesses of auto-regressive models with planning by introducing the notion of subgoal imbalanced or the idea that the true data distribution may not follow an autoregressive pattern. Using the sub-goal learning as a motivation, the authors propose multi-granularity diffusion model which uses a re-weighting procedure for training efficiency. Through a variety of planning tasks including Countdown, Sudoku and the boolean satisfiability problem (SAT), the authors demonstrate both the efficiency of the multi-granularity diffusion model compared to auto-regressive models and much better performance on more difficult tasks.

**Strengths:**

- Novelty: This work is particularly insightful. Not only does the work show experimentally demonstrate the success of diffusion models, it also provides intuition through the sub-goal problem. As mentioned in the related work, diffusion models have been explored in other domains as alternatives to auto-regressive models but it has not been done so in the planning domain. Given how frequently auto-regressive models are used, using diffusion models successfully for planning is almost by definition novel.
- Presentation and Clarity:  Related to the point above, by incorporating experiments through  early in the paper and providing intuition, the presentation made it much easier to understand.
- Contribution: There are several strong contributions. First, is replacing auto-regressive models with diffusion models in general. The second is the introduction of the multi-granularity diffusion model. The third, is showing that diffusion models are more efficient (especially with regard to tokens and on cheaper hardware, V100) for planning tasks.

**Weaknesses:**

- The main weakness would be on the types of planning tasks that the model is evaluated on. Other works like Tree of Thought evaluate on other planning tasks such as Mini-Crossword (in addition to the game of 24). It would be interesting to see how MDM does on tasks that also require some ‘common sense.’
- Scaling of MDM: Part of the advantages of AR models is the ability to scale up due to the ease of generating data with next-token prediction. Could MDMs exhibit such a property? In the results of the CD task, we see that with CD 4 and CD 5, the 85 M does better than the 303 M.

**Questions:**

In addition to the questions in the weaknesses:
- Could MDM exhibit the properties of in-context learning that AR models are known to have? Could it be possible once there is a sufficient amount of data?

**Details Of Ethics Concerns:**

In addition to the questions in the weaknesses:
- Could MDM exhibit the properties of in-context learning that AR models are known to have? Could it be possible once there is a sufficient amount of data?

---

> ### Author Response · Authors · 2024-11-22
>
> We sincerely thank Reviewer aoA3 for your review and are grateful for the time you spent on our submission. We are also glad you think our methodology is novel and has strong contributions. Below we would like to give detailed responses to each of your comments.
>
> **Weakness 1: The main weakness would be on the types of planning tasks that the model is evaluated on. Other works like Tree of Thought evaluate on other planning tasks such as Mini-Crossword (in addition to the game of 24). It would be interesting to see how MDM does on tasks that also require some ‘common sense.’**
>
> Thanks for the great suggestion. The research on “common sense” often needs to scale up to larger models with more data, while in our current study, we began with small-scale, controlled tasks to thoroughly analyze and validate the effectiveness of our approach. We acknowledge the significance of scaling up to larger models and more complex and common sense-related tasks. As a next step, we are actively working on extending our model to handle more complex planning tasks with scaled models. We believe this will further showcase the capabilities of our approach and its potential impact on a broader range of applications.
>
>
> **Weakness 2 Scaling of MDM: Part of the advantages of AR models is the ability to scale up due to the ease of generating data with next-token prediction. Could MDMs exhibit such a property? In the results of the CD task, we see that with CD 4 and CD 5, the 85 M does better than the 303 M.**
>
> Thanks for the great question. In our experiments, we compared models of different sizes (e.g., 85M and 303M parameters) using the same amount of training data for a strict comparison. The larger 303M model didn't outperform the 85M model on CD 4 and CD 5 tasks likely because the data was insufficient for the larger model to fully leverage its capacity, potentially leading to overfitting. This phenomenon also occurs in AR models under limited data conditions. We conducted additional experiments with increased data below, which shows that both increasing the model and data size leads to a consistent increase in performance on CD4 for both AR and MDM.
> We have included the experiment in the revised manuscript in Appendix D.3.
> |  | AR | MDM |
> |---|---|---|
> | 85M model, 500k data | 45.8 | 91.5 |
> | 303M model, 500k data | 41.3 | 88.3 |
> | 303M model, 1M data | 53.3 | 95.6 |
>
> **Question 1 Could MDM exhibit the properties of in-context learning that AR models are known to have? Could it be possible once there is a sufficient amount of data?**
>
> Thank you for the insightful question. We believe that with sufficient data and scaling, MDM could exhibit in-context learning properties similar to those of AR models, and we plan to investigate this in our future work.

---

> > ### Author Response · Authors · 2024-11-25
> >
> > Dear Reviewer,
> >
> > We would like to first express our sincere gratitude for your time and effort in reviewing our paper, and we truly appreciate your constructive feedback regarding our work.
> >
> > As the author-reviewer discussion period is coming to a close, we wonder if you could kindly share some of your thoughts so we can keep the discussion rolling to address your concerns if there are any. We are eager to address any additional questions or issues you may have.
> >
> > Thank you once again for your constructive feedback!
> >
> > Sincerely,
> >
> > The Authors

---

> > > ### Comment · Reviewer_aoA3 · 2024-11-26
> > >
> > > Thank you to the authors for addressing my questions. I keep the score at 8.

---

> > > > ### Author Response · Authors · 2024-11-26
> > > >
> > > > Thank you very much for your feedback! We greatly appreciate your comments and suggestions!

---

### Official Review · Reviewer_SUNs · 2024-11-05

**Soundness:** 2
**Presentation:** 4
**Contribution:** 3
**Rating:** 5
**Confidence:** 3

**Summary:**

This paper explores a few popular hard/challenging planning-based reasoning tasks such as Countdown and Sudoku, and whether a discrete diffusion model can solve them better than an autoregressive model. I am fairly unfamiliar with diffusion models and unable to judge whether the proposed method (multi-granularity diffusion modeling) is a significant contribution or a very minor one -- judged by Eq-8, it is a form of loss reweighting. The derivation for Eq (6) seems novel, but again, I am not fully sure if the derivation is a trivial reuse of previous work or a fully novel insight.

I have a few questions about several of the paper's claims. If properly addressed, I think this would be a good paper for acceptance.

**Strengths:**

The paper is very well-motivated and well-written.

I especially enjoyed Section 3, where the authors tried to use a simple metric (planning distance) to capture the difference between several methods (AR, Reverse-AR, and Diffusion). This alone is already a novel contribution.

The experimental result is mind-blowing. The proposed method almost completely solves Countdown (CD) and Game of 24, both are major challenges and are under active investigation. It would have been nice if, as reviewers, we had the time, energy, and resources to reproduce and replicate the results of the paper so that we could validate the claim. The authors have provided the code. A quick glance did not raise immediate red flags for me.

**Weaknesses:**

I do not consider these weaknesses, but any paper can always improve:

The datasets are mostly Countdown, Game of 24, and Sudoku. I think they are sufficient as of right now to show proof of concept. It would obviously be nicer if we saw other larger domains (such as Blocksworld [1]). However, the three datasets/domains investigated are the most popular ones and, therefore, sufficient.

[1] Valmeekam, Karthik, Sarath Sreedharan, Matthew Marquez, Alberto Olmo, and Subbarao Kambhampati. "On the planning abilities of large language models (a critical investigation with a proposed benchmark)." arXiv preprint arXiv:2302.06706 (2023).

**Questions:**

1. I want to understand Proposition 1 better: for a sequence $q(x)$, let's take your toy planning task as an example; if that is my true distribution, then I can just do an MC sample from $q(x)$ (by conditioning on previous states). Why would it not follow an autoregressive pattern? Is Proposition 1 as straightforward as thinking because there is an action policy $\pi$, therefore, $p_{\pi_1}(x_n | x_{1:n-1})) \not = p_{\pi_2}(x_n | x_{1:n-1}))$? I fail to see why this is the case.
2. "The regretful compromise" -- I fail to see what the compromise is. Can you tell me if this is you reinventing the concept of "compounding error" in [1]? If so, I am astonished that we seem to be recycling and relearning the same phenomenon every 5-7 years.
3. Looking at Eq (8), which is a per-token loss reweighting -- it seems that we can apply per-token loss reweighting, too. Can the authors comment on whether per-token loss reweighting can help AR, too (address the issue discussed in Proposition 1)?

[1] Lamb, Alex M., Anirudh Goyal ALIAS PARTH GOYAL, Ying Zhang, Saizheng Zhang, Aaron C. Courville, and Yoshua Bengio. "Professor forcing: A new algorithm for training recurrent networks." Advances in neural information processing systems 29 (2016).

---

> ### Author Response · Authors · 2024-11-22
>
> We sincerely thank Reviewer SUNs for your review and are grateful for the time you spent on our submission. We are happy to know you think our paper is well-motivated and you enjoy Section 3 especially. Below we would like to give detailed responses to each of your comments.
>
> **Weakness1 “ It would obviously be nicer if we saw other larger domains (such as Blocksworld [1]). However, the three datasets/domains investigated are the most popular ones and, therefore, sufficient.”**
>
> Thank you for your constructive feedback and for recognizing that the three datasets/domains we investigated are among the most popular and sufficient for demonstrating our approach. We agree that applying our method to larger domains such as Blocksworld [1] would further showcase its scalability and general applicability.
> Due to time constraints and computational resources, we were unable to include experiments on larger domains in this version of the paper. However, we will explore these domains in our future work.
>
> **Question 1: For a sequence $q(x)$, let's take your toy planning task as an example; if that is my true distribution, then I can just do an MC sample from $q(x)$ (by conditioning on previous states). Why would it not follow an autoregressive pattern?**
>
> Thank you for your insightful question. In the case of Figure 1, when the given goal is 9, the value 5 is not simply generated through random sampling; rather, it must be inferred by reasoning backward from the goal of 9. This backward inference is challenging for AR as it only utilizes the preceding context for its predictions. Broadly speaking, we would like to point out that there are some subgoal imbalance issues at the data level, especially in complex planning tasks where a certain action/token is determinative to the subsequent action/tokens. We show such a subgoal imbalance issue can be very challenging for AR. We have improved the description for Proposition 1 in the updated version.
>
>
> **Question2 "The regretful compromise" -- I fail to see what the compromise is. Can you tell me if this is you reinventing the concept of "compounding error" in [1]? If so, I am astonished that we seem to be recycling and relearning the same phenomenon every 5-7 years.**
>
> Thank you for your question. "Regretful compromise" here highlights a phenomenon: despite correctly calculating earlier steps, the model's incorrect planning forces it to make an incorrect calculation or decision at the final step in an attempt to achieve the overall goal. The model has the computational ability to find the correct solution, but because it cannot backtrack, it ends up making a mistake in trying to reach the goal. For example, in our paper, given the input “16, 4, 40, 51” and target 87, the prediction of AR is “51-40=11,16*4=64,11+64=87” while the correct solution is “16/4=4,40+51=91,91-4=87. The first two equations are calculated correctly, but the last one is incorrectly forced to equal 87. As shown in Figure 6(b), in many cases, the error rate for the last equation in AR (48.9%) is significantly higher than that of the previous equations (0.2% for the first and 7.2% for the second). This indicates that the model's lack of planning capability results in its realization of being unable to achieve the goal only at the end. By that point, it cannot correct the previous errors, leading it “regretfully” to rely on incorrect equations to reach the goal (e.g., the number 87 in the above case).
>
> "Compounding error" refers to the accumulation of small prediction errors over sequential steps, leading to a divergence from the expected trajectory, so it is more like a **general** cause, whereas "regretful compromise" here is the **specific** phenomenon we observe specifically in the goal-conditioned planning tasks when the model, unable to backtrack, makes a compromise at the final step due to the incorrect intermediate planning.

---

> > ### Author Response · Authors · 2024-11-22
> >
> > **Question3: Looking at Eq (8), which is a per-token loss reweighting -- it seems that we can apply per-token loss reweighting, too. Can the authors comment on whether per-token loss reweighting can help AR, too (address the issue discussed in Proposition 1)?**
> >
> > Thank you for raising this point. Per-token loss reweighting is indeed a technique that can be applied to AR models. However in our experiments, applying token reweighting to AR models still cannot solve subgoals with PD>1 (accuracy around 50%), similar to the original AR.
> > The core issues highlighted in Proposition 1 still exist in AR with token reweighting:
> > - **Lack of planning for future tokens**: Due to causal masking, AR models generate each token based solely on preceding tokens. Even with reweighted losses, this limitation prevents them from considering future context, which is essential for effective planning in tasks with token dependencies.
> > - **Inability to revise past tokens**: In tasks where tokens are interdependent, adjusting one token may affect others. AR models struggle with this because they cannot revisit and modify past outputs, whereas diffusion models can iteratively refine the entire sequence, allowing for adjustments across all tokens.
> >
> > We have included the experiment in the revised manuscript in Appendix D.2.
> >
> > | Planning Distance | AR | AR with token reweighting |
> > |---|---|---|
> > | 0 | 100 | 100 |
> > | 1 | 100 | 100 |
> > | 2 | 51.1 | 52.1 |
> > | 3 | 46.9 | 51.5 |
> > | 4 | 52.0 | 50.3 |
> > | 5 | 49.9 | 51.9 |

---

> > > ### Author Response · Authors · 2024-11-25
> > >
> > > Dear Reviewer,
> > >
> > > We would like to first express our sincere gratitude for your time and effort in reviewing our paper, and we truly appreciate your constructive feedback regarding our work.
> > >
> > > As the author-reviewer discussion period is coming to a close, we wonder if you could kindly share some of your thoughts so we can keep the discussion rolling to address your concerns if there are any. We are eager to address any additional questions or issues you may have.
> > >
> > > We would greatly appreciate it if you would consider adjusting the score based on our responses and the other review comments.
> > >
> > > Thank you once again for your constructive feedback!
> > >
> > > Sincerely,
> > >
> > > The Authors

---

> > > > ### Author Response · Authors · 2024-12-02
> > > > **Looking forward to further discussion**
> > > >
> > > > Dear Reveiwer SUNs,
> > > >
> > > > We would like to thank you again for your detailed reviews. Since the rebuttal deadline is approaching soon (Dec 2 AoE), we would appreciate it if you could kindly share some of your thoughts so we can keep the discussion rolling to address your concerns if there are any.
> > > >
> > > > For your reference, we make a summary of our paper's contribution:
> > > > - The insights on why planning is hard for AR through the lens of subgoal imbalance
> > > > - The design of minimal planning task to validate the shortcomings of AR
> > > > - The explanation of how diffusion learns subgoals that challenge the AR model both theoretically and empirically
> > > > - The proposing of Multi-granularity Diffusion Modeling (MDM), and showing it significantly outperforms autoregressive models on more complex reasoning challenges
> > > >
> > > > As most of the current generation of LLMs are AR-based, our work highlights the potential of diffusion-based approaches in advancing AI capabilities for sophisticated language understanding and problem-solving tasks, and we hope to promote the development of the next generation of diffusion-based LLMs.
> > > >
> > > > Thanks a lot for your time! Looking forward to your reply.
> > > >
> > > > Best regards,
> > > >
> > > > Authors

---

> > ### Comment · Reviewer_SUNs · 2024-12-03
> >
> > Thank you for responding with a detailed rebuttal. I'm not entirely satisfactory -- and I think the rebuttal raises further concerns/doubts.
> >
> > > Subgoal imbalance in autoregressive modeling...the generation of individual tokens may not inherently follow an autoregressive pattern...
> >
> > I think the updated Proposition 1 makes sense now. However, this statement does not make sense. The toy model defines a Markov Chain. In Markov Chain sampling, even if $p(5|7) = p(0|7)$, meaning the transition probability of going to $5$ of $7$ is equal, it still can be modeled as an autoregressive process. The rebuttal you guys wrote is saying that if there's backtracking, then the auto-regressive sampling would not capture the backtracking behavior. Backtracking comes from an algorithm you implemented to search through this Markov chain -- of course; it won't match the distribution of direct sampling from the Markov Chain. The way this statement is written is still very confusing. Please find a co-author or some people in your circle with a more rigorous background in sampling to rewrite this statement.
> >
> > >  This indicates that the model's lack of planning capability results in its realization of being unable to achieve the goal only at the end. By that point, it cannot correct the previous errors, leading it “regretfully” to rely on incorrect equations to reach the goal (e.g., the number 87 in the above case).
> >
> > I see -- the regret comes from realizing it should have made the decision differently. I accept this explanation. Please update the paper accordingly.
> >
> > > Q3
> >
> > I appreciate the additional experiment. It is satisfactory!
> >
> > > Given the true data distribution q(x) is unknown, the generation of individual tokens may not inherently follow an autoregressive pattern ((i.e., $x_n ∼ q(x_n | x_{1:n−1})$)
> >
> > This statement is still very unclear to me. Yes we do not have direct access to $q(x)$, but we are able to sample from it using a backtracking algorithm. The algorithm produces traces $\tilde q(x)$ and our AR model $p_{\theta}$ learns to model it. You are claiming $q(x) \not \sim \tilde q(x)$, which makes sense but is not what you wrote in the text. Please let me know exactly what you guys mean and what you guys plan to include in the final paper in the comment. I cannot recommend acceptance without knowing the paper has no technical error.

---

> > > ### Author Response · Authors · 2024-12-03
> > >
> > > Thank you very much for your feedback!
> > >
> > > **Q1: About "...the generation of individual tokens may not inherently follow an autoregressive pattern..."**
> > >
> > > Thanks for pointing out this potential confusion. We will update as follows:
> > > >Proposition 1: Subgoal imbalance due to the unknown data distribution $q(x)$}
> > > Given the data $x$ sampled from an unknown data distribution $q(x)$, the difficulty of learning each subgoal $x_n$ can differ significantly based on how we parametrize the model distribution, and some subgoals may require substantially more data to learn or may even be infeasible to learn.
> > >
> > > > Subgoal imbalance in autoregressive modeling.
> > > Given the data $x$ sampled from an unknown data distribution $q(x)$, the naive autoregressive modeling parametrizes the model distribution $p_\theta(x)$ as $p_\theta(x_1)\prod_{n=2}^{N}p_\theta(x_n \mid x_{1:n-1})$, the difficulty of learning each subgoal $x_n$ can differ significantly as given only the left context, and some subgoals may require substantially more data to learn or may even be infeasible to learn.
> > >
> > > We make an emphasis on the premise **given the data $x$ sampled from an unknown data distribution**, and it may be difficult for the AR models to learn given the data $x$. For example, some subgoals may depend on the backtrack trace as you also mentioned, and since the data is given, we cannot guarantee that all subgoals are conditioned on this trace. In this case, it becomes challenging for AR to learn effectively.
> > >
> > > Of course, we could construct $\tilde q(x)$ from $q(x)$ by adding the trace from a backtracking algorithm for learning in this case, but that is another topic (which would also introduce new issues, such as the learning distribution being different $q(x) \nsim \tilde q(x)$, how to choose the backtracking algorithm to construct the trace, and what to do if the trace is too long, etc.). **Here, we are discussing the modeling capabilities of AR and diffusion in the context where the given data is directly from the same true data distribution $q(x)$.**
> > >
> > > Additionally, we reported the performance of using AR to learn $\tilde q(x)$ from the Stream-of-Search paper in Table 1. We found that training with search trajectory supervision does provide some benefits, but its effectiveness is limited and significantly lags behind diffusion-based approaches (Lines 337-339).
> > >
> > > **Q2: I see -- the regret comes from realizing it should have made the decision differently. I accept this explanation. Please update the paper accordingly.**
> > >
> > > Thanks for the feedback. We will update Lines 468-471 as follows:
> > > > The first two equations are calculated correctly, but the last one is incorrectly forced to equal 87. Moreover, in many cases, the error rate for the last equation in AR (48.9\%) is significantly higher than that of the previous equations (0.2\% for the first and 7.2\% for the second). This indicates that the model's lack of planning capability results in its realization of being unable to achieve the goal only at the end. By that point, it cannot correct the previous errors, leading it “regretfully” to rely on incorrect equations to reach the goal (e.g., the number 87 in the above case).
> > >
> > > **Q3: Given the true data distribution q(x) is unknown, the generation of individual tokens may not inherently follow an autoregressive pattern**
> > >
> > > Please refer to Q1.
> > >
> > > Thank you again for your valuable feedback on improving our work. We hope our response could address your questions. If you have any further questions, we are happy to discuss them!

---

### Official Review · Reviewer_w8tZ · 2024-11-08

**Soundness:** 3
**Presentation:** 3
**Contribution:** 2
**Rating:** 6
**Confidence:** 4

**Summary:**

This paper introduces Multi-granularity Diffusion Modeling (MDM), a method that prioritizes subgoals by difficulty to improve complex reasoning and planning in tasks where autoregressive models fall short. MDM significantly outperforms autoregressive approaches on challenging tasks like Countdown and Sudoku, achieving accuracy rates of 91.5% and 100%, respectively. The results demonstrate the promise of diffusion-based methods for advancing AI in complex language understanding and problem-solving.

**Strengths:**

- The authors address a novel and important problem in reinforcement learning (RL): subgoal imbalance in planning tasks. This issue is particularly relevant for the application of deep learning models in RL.
- They propose a new method to tackle subgoal imbalance, which appears to be a reasonable approach to addressing this challenge.
- The authors demonstrate that their method outperforms traditional autoregressive models on complex tasks, including Countdown, Sudoku, and Boolean Satisfiability Problems.

**Weaknesses:**

- The evaluation is limited to well-designed, controlled tasks, which may restrict the general applicability of the proposed method. It would be interesting to see if this approach is also practical in more realistic RL scenarios.
- While they extend a diffusion model to handle subgoal imbalance, I wonder if a similar technique could be applied to autoregressive language models.  Extending this work to LLMs could make it more broadly applicable and impactful.
- The authors provide a strong example of subgoal imbalance in Section 3. However, it would be helpful to understand how frequently this issue occurs in real-world RL settings. Including concrete, real-world examples would enhance the appeal of this work.

**Questions:**

1. How does subgoal imbalance impact performance in real-world RL tasks? Could you provide concrete examples of this issue in practical applications?
2. How does the performance of the proposed method compare to autoregressive models in terms of computational efficiency (i.e., training time and inference time)?

---

> ### Author Response · Authors · 2024-11-22
>
> We sincerely thank Reviewer w8tZ for the review and are grateful for the time you spent with our submission. We wish to address your confusion and concerns by providing detailed responses to each of your comments.
>
> We would like to make our main point again first. Our starting point is to explore why current LLMs in the text field struggle with complex reasoning and long-term planning tasks. To this end, we made four contributions:
> - **The insights on why planning is hard, typically for AR, through the lens of subgoal imbalance**: We first point out the phenomenon of subgoal imbalance often present in complex reasoning and planning data, which makes it very difficult for the autoregressive (AR) structure of LLMs to learn certain subgoals (i.e., tokens), leading to poor prediction performance.
> - **The design of Minimal planning task**: We designed a minimal planning task to describe the concept of planning and experimentally validated the shortcomings of the AR structure.
> - **The explanation of how diffusion learns subgoals that challenge the AR model both theoretically and empirically**: We theoretically explain why diffusion methods can better address the subgoal imbalance issue and validate this on the aforementioned planning task.
> - **The propose of MDM**: We propose MDM to enhance the efficiency of existing diffusion models in optimization, which prioritizes subgoals based on difficulty during learning. We have validated that MDM significantly outperforms autoregressive models on more complex real-world tasks such as Countdown, Sudoku, and Boolean Satisfiability Problems.
>
> Most of the current generation of LLMs are AR-based. Our work highlights the potential of diffusion-based approaches in advancing AI capabilities for sophisticated language understanding and problem-solving tasks, and we hope to promote the development of the next generation of diffusion-based LLMs.
>
> **Weakness 1: The evaluation is limited to well-designed, controlled tasks, which may restrict the general applicability of the proposed method. It would be interesting to see if this approach is also practical in more realistic RL scenarios.**
>
> Thanks for the feedback. Since our focus is on exploring the issues with LLMs, we primarily concentrate on the text field. Extending the ideas and methods presented in this paper to an RL setting is interesting and feasible, but it falls outside the scope of our work.
>
> **Weakness 2:  While they extend a diffusion model to handle subgoal imbalance, I wonder if a similar technique could be applied to autoregressive language models. Extending this work to LLMs could make it more broadly applicable and impactful.**
>
> Thank you for your comment. The “subgoal” studied in this work particularly refers to “token” in language. Our work specifically addresses challenges in autoregressive (AR) language models when dealing with subgoal imbalance, and we find that diffusion models naturally offer a solution.
> The token reweighting technique can be applied to AR models as well, but it doesn’t overcome the inherent limitations of AR—namely, their inability to plan for future subgoals or revise past outputs based on future context. In our experiments, applying token reweighting to AR models still cannot solve subgoals with PD>1 (accuracy around 50%), similar to that in the original AR.
>
> | Planning Distance | AR | AR with token reweighting |
> |---|---|---|
> | 0 | 100 | 100 |
> | 1 | 100 | 100 |
> | 2 | 51.1 | 52.1 |
> | 3 | 46.9 | 51.5 |
> | 4 | 52.0 | 50.3 |
> | 5 | 49.9 | 51.9 |
>
> **Weakness 3: The authors provide a strong example of subgoal imbalance in Section 3. However, it would be helpful to understand how frequently this issue occurs in real-world RL settings. Including concrete, real-world examples would enhance the appeal of this work.**
>
> Thank you for your feedback. Our work primarily focuses on analyzing the shortcomings of AR language models in handling subgoal imbalance, which is a practical issue observed in existing AR LLMs, especially when dealing with complex reasoning and planning tasks. While subgoal imbalance may also impact the RL field, our main interest is in addressing this problem within the context of language modeling tasks where AR models face challenges in planning and reasoning due to their sequential generation process. Therefore, in this paper, we concentrate on scenarios relevant to AR LLMs rather than RL, and highlight the practical implications of subgoal imbalance in current language modeling applications.

---

> > ### Author Response · Authors · 2024-11-22
> >
> > **Question 1: How does subgoal imbalance impact performance in real-world RL tasks? Could you provide concrete examples of this issue in practical applications?**
> >
> > Please refer to Weakness3.
> >
> > **Question 2: How does the performance of the proposed method compare to autoregressive models in terms of computational efficiency (i.e., training time and inference time)?**
> >
> > Thank you for your question. In our experiments, we observed that the time for each training step is similar for both AR and MDM, primarily because the additional operations in MDM compared to AR are negligible (e.g., converting the causal mask to a full mask). Therefore, the training time is mainly related to the number of training steps. As shown in Figure 5, diffusion models often require longer training times for additional performance improvements, whereas AR models converge more quickly and no longer show performance gains.
> >
> > The inference time comparison of AR and MDM is shown in Figure 6(a), where we show samples per second on CD4 and CD5. For MDM, we can flexibly control the trade-off between accuracy and decoding speed by varying the diffusion timesteps T, achieving a faster speed than AR while maintaining better performance. For example, the samples per second on a single GPU for both AR and MDM on CD5 task are presented below.
> >
> > |  | Acc | Samples per second |
> > |---|---|---|
> > | AR | 5.1 | 2.2 |
> > | MDM T=1 | 12.7 | 22.5 |
> > | MDM T=2 | 24.5 | 19 |
> > | MDM T=4 | 35.5 | 13.1 |
> > | MDM T=8 | 39.8 | 7.5 |
> > | MDM T=16 | 44.5 | 4.1 |
> > | MDM T=32 | 45.1 | 2.3 |

---

> ### Author Response · Authors · 2024-11-25
>
> Dear Reviewer,
>
> We would like to first express our sincere gratitude for your time and effort in reviewing our paper, and we truly appreciate your constructive feedback regarding our work.
>
> As the author-reviewer discussion period is coming to a close, we wonder if you could kindly share some of your thoughts so we can keep the discussion rolling to address your concerns if there are any. We are eager to address any additional questions or issues you may have.
>
> We would greatly appreciate it if you would consider adjusting the score based on our responses and the other review comments.
>
> Thank you once again for your constructive feedback!
>
> Sincerely,
>
> The Authors

---

> > ### Author Response · Authors · 2024-12-02
> > **Looking forward to further discussion**
> >
> > Dear Reveiwer w8tZ,
> >
> > We would like to thank you again for your detailed reviews. Since the rebuttal deadline is approaching soon (Dec 2 AoE), we would appreciate it if you could kindly share some of your thoughts so we can keep the discussion rolling to address your concerns if there are any.
> >
> > For your reference, we make a summary of our paper's contribution:
> > - The insights on why planning is hard for AR through the lens of subgoal imbalance
> > - The design of minimal planning task to validate the shortcomings of AR
> > - The explanation of how diffusion learns subgoals that challenge the AR model both theoretically and empirically
> > - The proposing of Multi-granularity Diffusion Modeling (MDM), and showing it significantly outperforms autoregressive models on more complex reasoning challenges
> >
> > As most of the current generation of LLMs are AR-based, our work highlights the potential of diffusion-based approaches in advancing AI capabilities for sophisticated language understanding and problem-solving tasks, and we hope to promote the development of the next generation of diffusion-based LLMs.
> >
> > Thanks a lot for your time! Looking forward to your reply.
> >
> > Best regards,
> >
> > Authors

---

### Author Response · Authors · 2024-11-22

We deeply appreciate your insightful feedback and are grateful for the time you spent with our submission. Here we provide the summary of the contributions of our paper and the updates to the manuscript based on your valuable suggestions.

**Summary of Contributions:**
- The insights on why planning is hard for AR through the lens of subgoal imbalance
- The design of minimal planning task to validate the shortcomings of AR
- The explanation of how diffusion learns subgoals that challenge the AR model both theoretically and empirically
- The proposing of Multi-granularity Diffusion Modeling (MDM), and showing it significantly outperforms autoregressive models on more complex reasoning challenges

**Summary of updates to the manuscript:**
- We have improved the description of Proposition 1.
- We have added more details about what the view represents in diffusion in Line 246 and how it connects to Section 3.3.
- We have added the AR with token reweighting experiments in Appendix D.2.
- We have added the result of scaling both data and model size in Appendix D.3 as an extension of Table 1.

Thank you again for your contributions to improving our work. We hope our response could address your questions, and we are happy to address any further concerns or queries.

---

### Meta-Review · Area_Chair_nGAW · 2024-12-24

**Metareview:**

This paper proposes a particular discrete diffusion model to address reasoning and planning. All reviewers appreciate the merit and novelty of its technical contribution. Furthermore, it establishes a well-presented and clear connection between the bottlenecks in planning and why traditional autoregressive models would struggle with them. In particular, the problem is subgoal imbalance. Still, the definition of “subgoal imbalance” could leverage more illustrative examples. The theory developed can be more elaborate. The experiment setting is solid with hard problems. More real-world-based baselines definitely are welcome. Overall, it is solid writing with clear technical contributions.

**Additional Comments On Reviewer Discussion:**

Reviewers mostly appreciate this paper's proposed method before the rebuttal. The concern mainly lies on the diversity of the experiments and the underdevelopment of some of the theoretical perspectives. The rebuttal addresses most concerns.

---

### Decision · Program_Chairs · 2025-01-22

Accept (Poster)